# Computational Hardness of Reinforcement Learning with Partial $q^\pi$-Realizability

**Shayan Karimi**
University of Alberta
Edmonton, Canada
skarimi3@ualberta.ca

**Xiaoqi Tan**
University of Alberta
Edmonton, Canada
xiaoqi.tan@ualberta.ca

## Abstract

This paper investigates the computational complexity of reinforcement learning within a novel linear function approximation regime, termed *partial $q^\pi$-realizability*. In this framework, the objective is to learn an $\epsilon$-optimal policy with respect to a predefined policy set $\Pi$, under the assumption that all value functions corresponding to policies in $\Pi$ are linearly realizable. This framework adopts assumptions that are weaker than those in the $q^\pi$-realizability setting yet stronger than those in the $q^*$-realizability setup. As a result, it provides a more practical model for reinforcement learning scenarios where function approximation naturally arise. We prove that learning an $\epsilon$-optimal policy in this newly defined setting is computationally hard. More specifically, we establish NP-hardness under a parameterized *greedy policy set* (i.e., argmax) and, further, show that—unless NP = RP—an exponential lower bound (exponential in feature vector dimension) holds when the policy set contains *softmax policies*, under the Randomized Exponential Time Hypothesis. Our hardness results mirror those obtained in the $q^*$-realizability settings, and suggest that computational difficulty persists even when the policy class $\Pi$ is expanded beyond the optimal policy, reinforcing the unbreakable nature of the computational hardness result regarding partial $q^\pi$-realizability under two important policy sets. To establish our negative result, our primary technical contribution is a reduction from two complexity problems, $\delta$-MAX-3SAT and $\delta$-MAX-3SAT($b$), to instances of our problem settings: GLINEAR-$\kappa$-RL (under the greedy policy set) and SLINEAR-$\kappa$-RL (under the softmax policy set), respectively. Our findings indicate that positive computational results are generally unattainable in the context of partial $q^\pi$-realizability, in sharp contrast to the $q^\pi$-realizability setting under a generative access model.

## 1 Introduction

In reinforcement learning (RL), the term "*efficiency*" often refers to *statistical efficiency*, which relates to the number of samples required for an algorithm to converge to a near-optimal solution. Another critical, though less frequently addressed perspective is *computational complexity*, which focuses on the computational demands of the algorithm. To address these challenges, function approximation techniques—either *linear* or *nonlinear*—are commonly employed. These approximation methods enable the estimation of the value function so that resulting complexity bounds become independent of the size of the state space [TVR97, SB18, Ber09]. Instead, complexity depends on approximation parameters, thereby offering a scalable and efficient framework for RL in large state spaces.

Prominent RL applications employing *nonlinear* value approximators include AlphaGo [SHM+16], AlphaZero [SHS+17], MuZero [SAH+20], and AlphaStar [MOS+23]. Despite their practical success, nonlinear approximators are challenging to analyze theoretically. In contrast, *linear* value approximators allow for more tractable theoretical analysis. Two primary settings for linear function approxima-

39th Conference on Neural Information Processing Systems (NeurIPS 2025).

tion are commonly studied. The first, known as the $q^*$-*realizability* setting [WAS21, WR16], involves approximating only the optimal value function as a linear function. This setting makes no assumptions regarding the linearity of value functions under non-optimal policies, rendering it the least restrictive assumption. The second setting, known as the $q^\pi$-*realizability* setting [DKWY20], assumes that value functions for all policies are linearly realizable—a significantly stronger condition. Under the $q^*$- and $v^*$-realizability assumptions, recent works have established quasi-polynomial [KLLM22] and exponential lower bounds [KLL$^+$23] on computational complexity. Under the stronger assumption of $q^\pi$-realizability, it is known that computationally efficient methods can be achieved with access to a generative model [YHAY$^+$22].

**Motivating Questions.** The aforementioned studies motivate the following intriguing question:

> *Q1: Can we still achieve positive results (in terms of computational efficiency)*
> *in the $q^\pi$-realizability setting with a restricted policy class $\Pi$?*

Alternatively, the above question can be posed in the reverse direction:

> *Q2: Can we break the hardness result (in terms of computational efficiency) for the $q^*$-realizability*
> *setting when considering a policy class $\Pi$ with $\{\pi^*\} \subsetneq \Pi$?*

To illustrate why answering the questions above is interesting, let us consider the spectrum of linear function approximation in RL, characterized by two endpoints. At one end of this spectrum, we have $q^*$-realizability, representing a relatively weak assumption. At the opposite end is $q^\pi$-realizability, a significantly stronger realizability condition. Addressing the two questions posed above thus provides a means of bridging the gap between these two extreme cases. Towards this end, we propose the concept of *partial $q^\pi$-realizability*, formally introduced in Section 3. Under this partial realizability concept, our problem setting assumes access to a specific class of policies $\Pi$ within which the action-value functions for all state-action pairs are linearly realizable.

From a practical perspective, we can also view this partial $q^\pi$-realizability setting through the lens of agnostic RL with linear function approximation [JLR$^+$23]. In agnostic RL, we are given a policy set $\Pi$, and the goal is to learn a policy $\pi$ that competes with the best policy in the given policy class $\Pi \subset \mathcal{A}^{\mathcal{S}}$, where $\mathcal{A}$ and $\mathcal{S}$ denote the action and state spaces, respectively. Analogously, the partial $q^\pi$-realizability setting incorporates the key concept of inaccessibility to the optimal value function representation, highlighting the relationship between the agnostic RL framework and our problem setting when linear function approximation is employed.

**Main Contributions.** In this paper, we introduce *partial $q^\pi$-realizability* that bridges the gap between midler ($q^*$-realizability) and stronger ($q^\pi$-realizability) assumptions by defining the subclass of realizable policy sets for linear value function approximation. Based on the concept of partial $q^\pi$-realizability, we obtain two main results. First, we define an instance of partial $q^\pi$-realizability in which the policy set $\Pi$ consists of *deterministic greedy policies*. We show that learning a near-optimal policy in this setting is NP-hard. This result provides a partial negative answer to the aforementioned question Q1—it is not possible to ensure computational efficiency when access is limited to only a subclass of linearly realizable policies. Second, we move beyond the deterministic greedy policy set and study partial $q^\pi$-realizability under a more general policy set based on *stochastic softmax policies*. Under the randomized exponential time hypothesis (rETH), we prove that this problem setting exhibits *exponential computational hardness*, mirroring the result for the $q^*$-realizable setting. This result again provides a partial negative answer to the aforementioned question Q2—the computational hardness remains unbreakable even when considering a policy set richer than the optimal policy $\{\pi^*\}$.

As a broader implication, our hardness results for the partial $q^\pi$-realizability setting also show the existence of computational challenges inherent in Agnostic RL [JLR$^+$23] within the linear function approximation context.

## 2  Preliminaries

**Markov Decision Process.** In reinforcement learning [SB18, BT96, Put94], an agent aims to maximize its expected total reward over time by making optimal decisions through interaction with its given environment. In our problem setting, the environment is modeled as a Markov Decision

Process (MDP) $M := \langle \mathcal{S}, \mathcal{A}, \mathcal{P}, R, H \rangle$, where $\mathcal{S}$ is a discrete set of state, $\mathcal{A}$ is a discrete set of actions, $\mathcal{P} : \mathcal{S} \times \mathcal{A} \to \Delta(\mathcal{S})$ is the stochastic transition probability function which maps each $(s, a) \in \mathcal{S} \times \mathcal{A}$ to probability distribution over states, and $R : \mathcal{S} \times \mathcal{A} \to \Delta([0, 1])$ is the stochastic reward function, where $\Delta(X)$ is the set of probability distributions supported on set $X$. When we take action $a$ in state $s$, we get sampled next state $s' \sim \mathcal{P}(s, a)$, and $R \sim R(s, a)$ is a stochastic reward received for taking action $a$ in state $s$. In the episodic setting, the agent interacts with the environment starting from an initial state $s_1$, where the subscript denotes the time step or stage. During an episode, at each step $h$, taking an action $a_h$ in state $s_h$ results in a deterministic transition to the next state $s_{h+1}$ and yields a reward $R_h$. The episode continues until the agent reaches a terminal state set $\mathcal{S}_T \subset \mathcal{S}$ or meets another predefined stopping condition. We denote by $\mathcal{S}_h$ the set of states reachable from the initial state $s_1$ after exactly $h$ steps. Without loss of generality, we assume the state space $\mathcal{S}$ is partitioned into $H$ disjoint subsets, such that $\mathcal{S} = \mathcal{S}_1 \cup \mathcal{S}_2 \cup \cdots \cup \mathcal{S}_H$ and $\mathcal{S}_i \cap \mathcal{S}_j = \emptyset$ for any distinct $i, j \in [H]$. Additionally, we consider deterministic stationary policies $\pi : \mathcal{S} \to \mathcal{A}$, mapping each state to a single action, as well as stochastic policies $\pi : \mathcal{S} \to \Delta(\mathcal{A})$, which map each state $s \in \mathcal{S}$ to a probability distribution over actions. The state-value function $v^\pi(s)$ and the action-value function $q^\pi(s, a)$ are defined for any $s \in \mathcal{S}$ and $(s, a) \in \mathcal{S} \times \mathcal{A}$ under any policy $\pi$ as follows:

$$v^\pi(s) := \mathbb{E}_{\pi, s} \left[ \sum_{h=1}^{H} R_t \mid s_1 = s \right], q^\pi(s, a) := \mathbb{E}_{\pi, s, a} \left[ \sum_{h=1}^{H} R_t \mid s_1 = s, a_1 = a \right].$$

The optimal policy $\pi^*$ yields the optimal state-value function $v^*(s)$ and action-value function $q^*(s, a)$, formally defined as follows:

$$v^*(s) = \sup_\pi v^\pi(s) = \sup_{a \in A} q^{\pi^*}(s, a), \ q^*(s, a) = \sup_\pi q^\pi(s, a) = R(s, a) + \sup_\pi v^\pi(s'),$$

where $s'$ is the next state after taking action $a$ in state $s$. Furthermore, the interaction protocol between the agent and the environment is based on the *generative model*. The generative model—often referred to as the random access model [Kak03, SWW+19, YW19]—represents the strongest form of agent-environment interaction. Under this model, the learner can query a simulator with any $(s, a) \in (\mathcal{S} \times \mathcal{A})$ to obtain a sample $(s', R)$, where $s' \sim \mathcal{P}(s, a)$ and $R \sim R(s, a)$. This allows access to any state-action pair, regardless of whether the state has been previously visited. Throughout this paper, we assume the agent has access to a generative model.

**Linear Function Approximations.** In the function approximation setting, given a policy $\pi$, the action-value function $q^\pi(s_h, a)$ for any $(s_h, a) \in \mathcal{S}_h \times \mathcal{A}$ can be represented using a feature mapping $\phi : \mathcal{S} \times \mathcal{A} \to \mathbb{R}^d$, which assigns a $d$-dimensional feature vector to each state-action pair. We assume a linear function approximation, meaning there exists a weight vector $\theta_h \in \mathbb{R}^d$ such that

$$q^\pi(s_h, a) = \langle \phi(s_h, a), \theta_h \rangle. \tag{1}$$

This formulation expresses the action-value function as the inner product between the feature representation and the weight vector. Furthermore, we provide additional details on the different realizability settings in Appendix A.

**3-SAT and Randomized Exponential Time Hypothesis.** In the remainder of this section, we provide the relevant background on computational complexity. Let $\varphi$ be a Boolean formula, where the terms $x_i$ and $\bar{x}_i$ are referred to as *literals*. A formula $\varphi$ is said to be in *Conjunctive Normal Form (CNF)* if it can be expressed as

$$\varphi = C_1 \wedge C_2 \wedge \cdots \wedge C_n,$$

where $C_i$'s are called clauses and consist of disjunctions (ORs) of literals. In general, a $k$-SAT formula refers to a CNF-SAT problem in which each clause contains exactly $k$ literals. For 3-SAT, an NP-complete problem [Coo71, Lev73], the *Randomized Exponential Time Hypothesis (rETH)* is defined as follows:

**Definition 2.1** (Randomized Exponential Time Hypothesis (rETH) [DHM+14])**.** *There exists a constant $c > 0$ such that no randomized algorithm can solve the 3-SAT problem with $v$ variables in $2^{cv}$ time, with an error probability of at most $\frac{1}{3}$.*

Definition 2.1 emphasizes on the fact that randomized algorithms cannot solve NP-complete problems in sub-exponential time. As we will see, we use the rETH to establish an exponential lower bound for partial $q^\pi$-realizability setting under the softmax policy class $\Pi^{sm}$.

# 3 Problem Statement and Main Results

We begin this section by introducing the concept of *partial $q^\pi$-realizability*, followed by formal descriptions of our problem settings under the *greedy policy set* $\Pi^g$ and the *softmax policy set* $\Pi^{sm}$. We then present our two main hardness results, stated in Theorem 3.1 (under $\Pi^g$) and Theorem 3.2 (under $\Pi^{sm}$).

## 3.1 Problem Statement and Definitions

Let us first define the concept of *partial $q^\pi$-realizability* in Definition 3.1 below.

**Definition 3.1** (**Partial $q^\pi$-realizability under $\Pi$**). *Given a policy set $\Pi \subset \mathcal{A}^{\mathcal{S}}$ and a feature vector $\phi : \mathcal{S} \times \mathcal{A} \to \mathbb{R}^d$, an MDP is said to be partially $q^\pi$-realizable under $\Pi$ if, for all $\pi \in \Pi$, there exists $\theta_h \in \mathbb{R}^d$ such that:*

$$q_h^\pi(s_h, a_h) = \langle \phi(s_h, a_h), \theta_h \rangle \quad \forall (s_h, a_h) \in \mathcal{S}_h \times \mathcal{A}.$$

Definition 3.1 ensures the exact linear realizability when using $\phi \in \mathbb{R}^d$ under a subset of policies $\Pi \subset \mathcal{A}^{\mathcal{S}}$. Clearly, partial $q^\pi$-realizability is a weaker assumption than the $q^\pi$-realizability assumption, since we only assume linear realizability under a given policy set instead of under all the possible policies. On the other hand, partial $q^\pi$-realizability is stronger than $q^*$-realizability since the policy set $\Pi$ may include the optimal policy. Thus, by Definition 3.1, we can bridge the gap between these two extreme cases of realizability assumptions. We refer to $\phi$ and $\theta$ as partial realizability *feature* and *weight* vectors, respectively.

**Definition 3.2** (**Learner's objective: $\epsilon$-optimality relative to $\Pi$**). *Given an initial state $s_1 \in \mathcal{S}$, a policy $\pi \in \mathcal{A}^{\mathcal{S}}$ is $\epsilon$-optimal with respect to the best policy in $\Pi$ if*

$$\max_{\hat{\pi} \in \Pi} v^{\hat{\pi}}(s_1) - v^\pi(s_1) \leq \epsilon. \tag{2}$$

*In the case of a learner that employs randomized algorithms, we return the policy $\pi$ that satisfies the above condition with high probability. Note that the learned policy $\pi$ is deterministic when the policy class $\Pi$ is deterministic, and stochastic when $\Pi$ is stochastic.*

**Greedy Policy Set $\Pi^g$.** As our first attempt to have a well-defined $\Pi$, we define a special class of policy set called *greedy policy set*, which is parameterized by a given feature vector $\phi' \in \mathbb{R}^{d'}$. The details regarding this parameterization are given in the definition below.

**Definition 3.3** (**Greedy policy set $\Pi^g$**). *Let $\phi' : \mathcal{S} \times \mathcal{A} \to \mathbb{R}^{d'}$ be a feature vector with dimension $d' \in \mathbb{N}$. For any $h \in [H]$ and $\theta' \in \mathbb{R}^{d'}$, let $\pi_{\theta'}^g : \mathcal{S}_h \to \mathcal{A}$ be defined as follows:*

$$\pi_{\theta'}^g(s_h) := \arg\max_{a \in A} \langle \phi'(s_h, a), \theta' \rangle \qquad \forall s_h \in \mathcal{S}_h. \tag{3}$$

*In the event of a tie in the argmax, the action with the lowest index is selected. The greedy policy set, induced by any $\theta' \in \mathbb{R}^{d'}$, is defined as:*

$$\Pi^g := \{\pi_{\theta'}^g | \theta' \in \mathbb{R}^{d'}\}. \tag{4}$$

By Definition 3.3, the action $\pi_{\theta'}^g(s_h)$ is greedy with respect to $\langle \phi'(s_h, a), \theta \rangle$ (i.e., argmax). Also note that changing $\theta'$ can result in different greedy actions at each state $s_h \in \mathcal{S}_h$. Thus, Definition 3.3 implies that for different values of $\theta'$, we may obtain different greedy actions $\pi_{\theta'}^g(s_h)$ for each $s_h \in \mathcal{S}_h$. We refer to $\phi'$ and $\theta'$ as the *policy set parametrization* (PSP) feature and weight vectors, respectively. We are now ready to formally define our first complexity problem:

---

**Complexity Problem**   GLINEAR-$\kappa$-RL

*Input:*       An MDP $M$ that is partially $q^\pi$-realizable under $\Pi^g$ with $\kappa$ actions, having access to feature vectors ($\phi \in \mathbb{R}^d$ and $\phi' \in \mathbb{R}^{d'}$), horizon $H = \Theta(d^{\frac{1}{3}})$, and state space of size $\exp(\Theta(d^{\frac{1}{3}}))$.

*Goal:*        Find a policy $\pi$ such that it satisfies the $\epsilon$-optimality condition in Equation (2).

---

In the complexity problem **GLINEAR-$\kappa$-RL**, "G" stands for greedy. As a special instance of GLINEAR-$\kappa$-RL, we define the two-action version of the GLINEAR-$\kappa$-RL problem as GLINEAR-2-RL.

**Softmax Policy Set $\Pi^{sm}$.** Softmax policies are widely used in RL, particularly in methods such as policy gradients [SMSM00, HZAL18, SWD$^+$17, SLM$^+$17] and actor-critic algorithms [KT99]. Their effectiveness in exploration and smooth parameterization makes softmax a natural choice in these settings. Let us first review the definition of the well-known softmax policies.

**Definition 3.4** (**Softmax Policy Set $\Pi^{sm}$**). *Let $\phi' : \mathcal{S} \times \mathcal{A} \to \mathbb{R}^{d'}$ be a feature vector with dimension $d' \in \mathbb{N}$. For any $h \in [H]$ and $\theta' \in \mathbb{R}^{d'}$, let $\pi_{\theta'} : \mathcal{S}_h \to \Delta(\mathcal{A})$ be defined as follows:*

$$\pi_{\theta'}(a|s_h) = \frac{e^{\phi'(s_h,a)^\intercal \theta'}}{\sum_{i=1}^{\kappa} e^{\phi'(s_h,a_i)^\intercal \theta'}}. \qquad \forall s_h \in \mathcal{S}_h. \tag{5}$$

*Here, $\kappa$ represents the total number of actions available in state $s_h \in \mathcal{S}_h$. The policy $\pi_{\theta'}(s_h)$ defines a probability distribution over the actions, which is updated as $\theta'$ changes. Note that we use the softmax without a temperature parameter. The softmax policy set, induced by any $\theta' \in \mathbb{R}^{d'}$, is defined as:*

$$\Pi^{sm} := \{\pi_{\theta'} | \theta' \in \mathbb{R}^{d'}\}. \tag{6}$$

Following the format of the complexity problem GLINEAR-$\kappa$-RL, here we define our second complexity problem, termed **SLINEAR-$\kappa$-RL**, which has the same objective as GLINEAR-$\kappa$-RL but under the softmax policy set $\Pi = \Pi^{sm}$. In the complexity problem SLINEAR-$\kappa$-RL, "S" stands for softmax. As a special instance of SLINEAR-$\kappa$-RL, we define the two-action version of the SLINEAR-$\kappa$-RL problem as SLINEAR-2-RL.

## 3.2 Main Results: Hardness of Solving GLINEAR-$\kappa$-RL and SLINEAR-$\kappa$-RL

We show that solving the complexity problems GLINEAR-$\kappa$-RL and SLINEAR-$\kappa$-RL in a computational efficient way is impossible. The formal result under $\Pi^g$ is given in Theorem 3.1 below.

**Theorem 3.1** (**NP-Hardness of GLINEAR-$\kappa$-RL**). *Solving the **GLINEAR-$\kappa$-RL** problem for $\epsilon \leq 0.05$ is NP-hard. More specifically, there exists no polynomial-time algorithm (in terms of the parameters $(d, d', H)$) to compute an $\epsilon$-optimal policy for the GLINEAR-$\kappa$-RL problem, unless $P = NP$.*

Theorem 3.1 demonstrates that achieving $\epsilon$-optimality for small values of $\epsilon$ in GLINEAR-2-RL is computationally intractable. Notably, Theorem 3.1 reveals that under a specified greedy policy set (Definition 3.3), the results diverge from those achievable in the computationally efficient $q^\pi$-realizability setting. On the other hand, Theorem 3.1 suggests that even with an expanded policy set, existing computational hardness results for the $q^*$-realizability setting remain unbroken. The main idea of the proof is to reduce an NP-hard problem to GLINEAR-$\kappa$-RL in polynomial time. We provide a full proof of Theorem 3.1 in the next section.

As our second contribution to the hardness result, we leverage the randomized exponential time hypothesis (Definition 2.1) to derive a computational lower bound for solving the SLINEAR-2-RL problem. In the softmax policy set regime, algorithms that interact with the MDP inherently involve internal randomness. Therefore, a standard NP-hardness result is insufficient, as it only rules out polynomial-time *deterministic* algorithms and does not capture the limitations of randomized methods. To address this gap, we strengthen our hardness result under the more stringent computational assumption of rETH, which goes beyond the classical NP $\neq$ P conjecture. Our main theorem regarding the hardness of the SLINEAR-2-RL problem is stated below.

**Theorem 3.2** (**Hardness under $\Pi^{sm}$**). *Under rETH, there exist some small constant $\epsilon_0$ such that for any $\epsilon \leq \epsilon_0$, no randomized algorithm can solve the **SLINEAR-$\kappa$-RL** problem in time $\exp\left(o\left(\frac{d^{\frac{1}{3}}}{\text{polylog}(d^{\frac{1}{3}})}\right)\right)$ with error probability $\frac{1}{10}$, where $d$ denotes the dimension of the partial realizability feature vector $\phi$.*

Theorem 3.2 establishes the computational hardness of achieving an $\epsilon$-optimal policy under partial realizability of the softmax policy class $\Pi^{sm}$. It is worth noting that, although $\Pi^g \subset \Pi^{sm}$ holds with

high probability, the hardness result for GLINEAR-$\kappa$-RL relies solely on the standard assumption that $\mathrm{NP} \neq \mathrm{P}$, whereas the stronger result for SLINEAR-$\kappa$-RL requires the more stringent randomized Exponential Time Hypothesis (rETH). An additional insight from Theorems 3.1 and 3.2 is that both GLINEAR-$\kappa$-RL and SLINEAR-$\kappa$-RL are NP-hard, revealing a subtle paradox: even when the policy class is infinite and linearly realizable, learning under partial $q^\pi$-realizability remains computationally intractable.

# 4 Proof Sketch of Theorem 3.1

In this section, we provide a high-level sketch of the proof of Theorem 3.1, outlining the core techniques and ideas used to establish the hardness result. The proof consists of two main components, which we describe at a conceptual level. Since the techniques developed for GLINEAR-2-RL readily extend to the SLINEAR-2-RL setting with only minor modifications, we defer the full proof of the hardness result under softmax policies to Appendix E.

## 4.1 An Overview of Our Proof

It is known that reduction procedures are essential for establishing computational lower bounds. Specifically, to prove that a problem $\mathcal{L}_1$ is NP-hard, we reduce a known NP-hard problem $\mathcal{L}_2$ to $\mathcal{L}_1$. In this context, we use the decision version of the MAX-3SAT problem, referred to as $\delta$-MAX-3SAT. The formal definition of MAX-3SAT is given as follows:

**Definition 4.1** (MAX-3SAT). *Given a 3-CNF Boolean formula $\varphi$ with $n$ variables, denoted by the set $\mathcal{X} = \{x_1, \ldots, x_n\}$, and $k$ clauses, denoted by the set $\mathcal{C} = \{c_1, c_2, \ldots, c_k\}$, the MAX-3SAT problem is to find an assignment that satisfies the maximum number of clauses in $\varphi$.*

Let $\mathcal{X}_{\text{assign}} = \{0, 1\}^n$ denote the set of all possible assignments to the variables in $\mathcal{X} = \{x_1, \ldots, x_n\}$. Each element $x \in \mathcal{X}_{\text{assign}}$ represents an assignment where $x_i \in \{0, 1\}$ gives the truth value of $x_i$ for $i = 1, \ldots, n$. For any $x \in \mathcal{X}_{\text{assign}}$, let $\mathcal{C}_{\text{true}}(x)$ and $\mathcal{C}_{\text{false}}(x)$ denote the set of clauses that are satisfied and unsatisfied, respectively. Clearly, $|\mathcal{C}_{\text{true}}(x)| + |\mathcal{C}_{\text{false}}(x)| = |\mathcal{C}|$ holds for all $x \in \mathcal{X}_{\text{assign}}$. The goal of MAX-3SAT is to find an assignment $x \in \mathcal{X}$ such that $|\mathcal{C}_{\text{true}}(x)|$ is maximized. We also define the decision version of MAX-3SAT as follows:

**Definition 4.2** ($\delta$-MAX-3SAT). *Given a MAX-3SAT problem $\varphi$ with $n$ variables, denoted by the set $\mathcal{X} = \{x_1, \ldots, x_n\}$, $\delta$-MAX-3SAT is defined as the following decision problem: output "Yes" if we can find an assignment $x \in \mathcal{X}_{assign}$ such that $\frac{|\mathcal{C}_{true}(x)|}{|\mathcal{C}|} \geq 1 - \delta$, i.e., at least $1 - \delta$ fraction of the clauses are satisfied; output "No" otherwise.*

The following lemma shows that for any $\delta < \frac{1}{8}$, solving $\delta$-MAX-3SAT is NP-hard.

**Lemma 4.1** (NP-hardness of $\delta$-MAX-3SAT [Hå01]). *It is NP-hard to approximate the MAX-3SAT problem within a factor of $\frac{7}{8}$. Specifically, unless P = NP, no polynomial-time algorithm can guarantee that more than $\frac{7}{8}$ of the maximum number of satisfiable clauses are satisfied for any given MAX-3SAT formula.*

The key idea of our proof is to reduce the $\delta$-MAX-3SAT problem to GLINEAR-2-RL. Through this reduction process, we show that a solution to GLINEAR-2-RL can effectively solve $\delta$-MAX-3SAT, indicating that the GLINEAR-2-RL problem is at least as hard as $\delta$-MAX-3SAT for some $\delta < \frac{1}{8}$. In the subsequent sections, our proof involves a two-action version of the GLINEAR-$\kappa$-RL problem, GLINEAR-2-RL, and shows that the GLINEAR-2-RL is NP-hard. More specifically, we denote the learner interacting with GLINEAR-2-RL as algorithm $\mathcal{A}_{\text{RL}}$. We aim to design an $\mathcal{A}_{\text{SAT}}$ algorithm—which interacts with $\delta$-MAX-3SAT—derived from $\mathcal{A}_{\text{RL}}$, an algorithm that interacts with a properly designed MDP $M_\varphi$ derived from the $\delta$-MAX-3SAT instance $\varphi$. Following this idea, we complete the proof of Theorem 3.1 by showing the following two steps:

**Step-1: Polynomial construction of $M_\varphi$.** We show that it is possible to design an MDP instance $M_\varphi$ (with the given attributes GLINEAR-2-RL) from a given $\delta$-MAX-3SAT instance $\varphi$ in polynomial time. This requires the polynomial-time construction of the partial realizability features and weight vectors, along with the polynomial-time construction of parametric policies. These steps form the first part of the proof, which is outlined in Section 4.2.

**Step-2: Algorithmic connection between $\mathcal{A}_{\text{SAT}}$ and $\mathcal{A}_{\text{RL}}$.** Recall that the learner interacting with GLINEAR-2-RL using algorithm $\mathcal{A}_{\text{RL}}$, and the algorithm solving $\delta$-MAX-3SAT is denoted as $\mathcal{A}_{\text{SAT}}$. We show that if $\mathcal{A}_{\text{RL}}$ succeeds on $M_\varphi$, then $\mathcal{A}_{\text{SAT}}$ can be algorithmically derived from $\mathcal{A}_{\text{RL}}$ to solve the $\delta$-MAX-3SAT instance $\varphi$ with comparable efficacy. Specifically, we show that if the $\delta$-MAX-3SAT instance $\varphi$ is $(1 - \delta + 2\epsilon)$-satisfiable for some $\epsilon \leq \frac{\delta}{2}$, and algorithm $\mathcal{A}_{\text{RL}}$ returns an $\epsilon$-optimal policy $\pi$ in solving the MDP instance $M_\varphi$, then following the policy $\pi$ for setting the value of variables in our $\varphi$, we can satisfy at least $1 - \delta$ fraction of clauses in the $\delta$-MAX-3SAT instance $\varphi$, leading to a contradiction and completing the hardness proof.

In the next section, we present the details of Step-1, as the key to our reduction lies in the successful polynomial-time construction of the MDP instance $M_\varphi$. The details of Step-2 are deferred to Appendix D.

## 4.2 An In-depth Look at Step-1: Polynomial Construction of $M_\varphi$

As outlined in Section 4.1, we must polynomially transform a given $\delta$-MAX-3SAT instance $\varphi$—with the attributes specified in Section 4.1—into a MDP instance that encapsulates the structural properties of the GLINEAR-2-RL problem. We assume the $\delta$-MAX-3SAT instance has $n$ variables and, without loss of generality, that the variables first appear in clauses in the order $(x_1, x_2, \ldots, x_n)$.

### 4.2.1 Design of the MDP Structure

We first formalize the structural design of the MDP instance derived from the $\delta$-MAX-3SAT problem, ensuring the attributes of the GLINEAR-2-RL problem are embedded within the MDP.

**States, Actions, and Transitions.** Each state is represented as a $n$-tuple of -1, 0, and 1, starting with the initial state given by $s_1 = (-1, -1, \ldots, -1)$. For any state $s_h \in \mathcal{S}_h$, we can take two possible actions $a_h = \{0, 1\}$, namely, the action set of the MDP is $\mathcal{A} = \{0, 1\}$. Once action $a_h$ is taken, the MDP will transit to the new state $s_{h+1}$, where $s_{h+1}$ mirrors the current state $s_h$ for all the elements except that the $h$-th element will be changed from -1 to $a_h$. For example, if we take action $a_1 = 0$ at stage $h = 1$, the next state will be $s_1 = (0, -1, \cdots, -1)$. In general, for any $h \in [H]$, the state $s_h$ can be represented as $s_h = (x_1, \cdots, x_{h-1}, -1, \cdots, -1)$, where $\{x_i\}_{\forall i \in [h-1]}$ denote the variables of the MAX-3SAT problem that have been assigned based on the actions taken in the first $h - 1$ stages. Therefore, the transition dynamics in GLINEAR-2-RL are deterministic. Namely, starting from any state $s_h = (x_1, \cdots, x_{h-1}, -1, \cdots, -1) \in \mathcal{S}_h$, selecting an action $a_h \in \mathcal{A}$ will lead to a deterministic transition to the next state $s_{h+1}$, defined as follows:

$$\Pr[s_{h+1} = (x_1, \cdots, x_{h-1}, 0, -1, \ldots, -1) | s_h, a_h = 0] = 1,$$
$$\Pr[s_{h+1} = (x_1, \cdots, x_{h-1}, 1, -1, \ldots, -1) | s_h, a_h = 1] = 1.$$

Note that, based on the structure of the action set $\mathcal{A}$ and the state space $\mathcal{S}$, our MDP has a binary tree structure with $2^{n+1} - 1$ states.

**Terminal States.** As mentioned earlier, GLINEAR-$\kappa$-RL is an episodic problem, and we are in a finite horizon setting. Let $\mathcal{S}_H$ denote the set of terminal states located at the final stage $H = n + 1$. Once one of these states is reached, the episode concludes. Note that for any state $s_H \in \mathcal{S}_H$, all the $n$ elements are assigned to be 0 or 1, depending on the actions $a_1, \cdots, a_{H-1}$.

**Rewards.** The reward of the MDP is zero at all stages except at the final stage $h = H$. For any final state $s_H \in \mathcal{S}_H$, the reward is defined as the ratio of the satisfied clauses to the total number of clauses. Namely, the reward $R(s_h)$ is defined as

$$R(s_h) = \begin{cases} \frac{|\mathcal{C}_{\text{true}}(s_h)|}{|\mathcal{C}|} & \text{if } h = H, \\ 0 & \text{otherwise,} \end{cases}$$

where $|\mathcal{C}|$ denotes the cardinality of the set of all the clauses in the $\delta$-MAX-3SAT problem $\varphi$, and $\mathcal{C}_{\text{true}}(s_H)$ denotes the set of satisfied clauses in $\varphi$ when the variables $x = (x_1, \cdots, x_n)$ are assigned in an element-wise manner to be the value of the final state $s_H$, i.e., $x = s_H$. To better illustrate our MDP design, we provide an example in Figure 4.2.1, with additional details presented in Example 4.1 below.

**Example 4.1.** *To better illustrate the constructed MDP, we give an example based on Fig. 4.2.1. Suppose that we are given a MAX-3SAT problem $\varphi$ as follows:*

$$\varphi : (x_1 \lor \bar{x}_2 \lor x_3) \land (\bar{x}_1 \lor x_2 \lor \bar{x}_3). \tag{7}$$

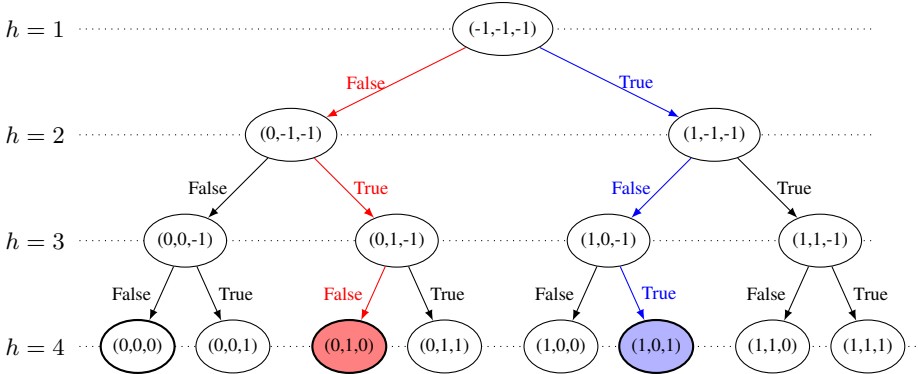

Figure 4.2.1: Given a MAX-3SAT problem $\varphi : (x_1 \vee \bar{x}_2 \vee x_3) \wedge (\bar{x}_1 \vee x_2 \vee \bar{x}_3)$ with $n = 3$ variables, the constructed MDP consists of 15 states, each represented as a $n$-tuple of -1, 0, and 1. The initial state is defined as $(-1, -1, -1)$ (i.e., the root of the tree). Actions are either "True" or "False," and the total horizon is $H = n + 1 = 4$. Starting from the initial state $(-1, -1, -1)$ at step $h = 1$, if the action "False" is taken (i.e., $a_1 = 0$), the first element of the next state is set to 0, resulting in the state $(0, -1, -1)$ at step $h = 2$ (the red path). Alternatively, if the action "True" is taken (i.e., $a_1 = 1$), the first element of the next state is set to 1, leading to the state $(1, -1, -1)$ at step $h = 2$ (the blue path). In the figure, both the red and blue paths yield a total reward of $\frac{1}{2}$.

*This* MAX-3SAT *formula is always satisfied except when* $(x_1, x_2, x_3) = (0, 1, 0)$ *or* $(1, 0, 1)$, *in these cases the ratio of clauses that can be satisfied is* $\frac{1}{2}$. *Here, state represent a tuple containing variables of our* MAX-3SAT *problem, and the actions are either "True" (1) or "False" (0). In this instance, we have 8 states as our terminal states (starting from* $(1, 1, 1)$ *in* $h = 4$). *Starting from the initial state* $(-1, -1, -1)$, *we have two actions: "True" or "False", which can transit us to the second state denoted as* $(1, -1, -1)$ *or* $(0, -1, -1)$. *For example, if action "True" is taken in state* $(-1, -1, -1)$, *we transit deterministically to state* $(1, -1, -1)$. *The final stage here is* $h = 4$. *To illustrate partial satisfiability in the* MAX-3SAT *problem instance* $\varphi$, *consider the red path starting from state* $(-1, -1, -1)$ *to state* $(0, 1, 0)$, *and the blue path from state* $(-1, -1, -1)$ *to state* $(1, 0, 1)$. *The total rewards of these two paths are* $\frac{1}{2}$ *(only one of the clauses is satisfied), while the total reward of all the other paths are* 1 *(all of the clauses are satisfied).*

### 4.2.2 Design of PSP Feature and Weight Vectors $\phi'$ and $\theta'$

To guarantee that our MDP captures the attributes of the GLINEAR-2-RL problem, we must ensure that PSP feature vector $\phi'$ is properly designed, allowing for the construction of well-defined greedy policy sets (Definition 3.3).

For any given state-action pair $(s_h, a) \in \mathcal{S}_h \times \mathcal{A}$, let $\phi' \in \mathbb{R}^{d'}$, where $d' = n$, be defined as follows:

$$\phi'(s_h, a) = \begin{cases} \left[ 0, 0, \cdots, \underbrace{1}_{h\text{-th element}}, 0, \cdots, 0 \right] & \text{if } a = \text{True}, \\[4mm] \left[ 0, 0, \cdots, \underbrace{-1}_{h\text{-th element}}, 0, \cdots, 0 \right] & \text{if } a = \text{False}, \end{cases} \tag{8}$$

where all the elements of $\phi'(s_h, a)$ are 0 except the $h$-th one, which is set to 1 (when $a =$"True") or $-1$ (when $a =$"False"). Recall that the action set $\mathcal{A} = \{0, 1\}$, where 0 corresponds to action "False", and 1 corresponds to action "True". For any $s_h$, if $\pi^g_{\theta'}(s_h) = 1$, then the action $a = 1$ is selected as the result of the maximization:

$$\arg\max_{a \in \mathcal{A}} \langle \phi(s_h, a), \theta' \rangle = 1. \tag{9}$$

In this case, it suffices to set the $h$-th element of the vector $\theta'$ as an arbitrary positive number. In contrast, if $a = -1$ needs to be selected as the result of the above argmax, we can set the $h$-th element of the vector $\theta'$ as an arbitrary negative number. Without loss of generality, we assume that $\theta' \in \mathbb{R}^{d'}$

consists only of elements in $\{-1, 1\}$. Note that for each given $\theta'$, the greedy actions selected in all the states in $\mathcal{S}_h$ are the same, depending on whether the $h$-th entry of $\theta'$ being 1 or -1.

### 4.2.3 Design of partial realizability Feature and Weight Vectors $\phi$ and $\theta$

As demonstrated in the complexity problem GLINEAR-$\kappa$-RL and Definition 3.1, the given partial realizability feature vector $\phi$ is utilized for the purpose of holding exact linear realizability under a given greedy policy set $\Pi^g$. In this section, we show how to design $\phi, \theta_h \in \mathbb{R}^d$ such that the action-value function $q_h^\pi(s_h, a_h)$ can be represented as a linear combination of $\phi$ and $\theta_h$ for any $(s_h, a_h) \in \mathcal{S}_h \times \mathcal{A}$ under any greedy policy $\pi \in \Pi^g$ for all $h \in [H]$. Formally, we state the linear realizability results as follows:

**Proposition 4.1.** *Given a greedy policy set $\Pi^g$, for any state $s_h \in \mathcal{S}_h$, $a_h \in \mathcal{A}$, and $\pi \in \Pi^g$, there exist $\phi(s_h, a) \in \mathbb{R}^d$ and $\theta_h \in \mathbb{R}^d$ such that:*

$$q_h^\pi(s_h, a_h) = \langle \phi(s_h, a_h), \theta_h \rangle, \qquad \forall h \in [H]. \tag{10}$$

*Proof.* We prove the above proposition in three steps. First, we design $\phi(s_h, a)$ to track the number of satisfied clauses in $\delta$-MAX-3SAT up to stage $h$, while also keeping a record of the remaining unsatisfied clauses after each variable assignment. Next, we construct the partial realizability weight vector $\theta_h$ to track the policy $\pi$ from stage $h + 1$ onward. We then show by induction that the linear realization described in Proposition 4.1 holds. See Appendix C for the details. □

Before concluding this section, we present two remarks: one on the polynomial-time construction of the MDP parameters in $M_\varphi$, and another on the conditions required for establishing NP-hardness of partial $q^\pi$-realizability under a general policy set $\Pi$.

**Remark 4.1** (Polynomial-Time Construction). *The computationally intensive step in our reduction involves designing the feature vectors $\phi$ and $\theta$, which can be done in $O(n^3)$ time due to their dependence on the size of $\mathcal{C}_{total}$. Other components such as $\phi'$, $\theta'$, and reward vectors require only $O(n)$ time. Thus, the entire reduction remains within polynomial time, satisfying the complexity-theoretic requirements of the NP-hardness proof. A step-by-step example is provided in Appendix B.*

**Remark 4.2** (General Conditions for NP-hardness under Partial $q^\pi$-realizability). *Our proof framework naturally extends to other policy classes. For any NP-hard decision problem $\mathfrak{P}$ with input size $|\mathfrak{P}|$ and any policy set $\Pi \subset \mathcal{A}^\mathcal{S}$ of size $\Omega(|\mathcal{S}|)$, learning an $\epsilon$-optimal policy under partial $q^\pi$-realizability is NP-hard, provided that (i) each policy $\pi \in \Pi$ can be parameterized in polynomial time in $|\mathfrak{P}|$, (ii) the feature vectors $\phi$ and $\theta$ are constructible in polynomial time, and (iii) solving $\mathfrak{P}$ reduces to finding an $\epsilon$-optimal policy with respect to $\Pi$. These conditions highlight the central role of efficient policy representation in the reduction.*

## 5 Conclusion and Future Work

In this paper, we introduced a novel linear realizability setting in reinforcement learning (RL), termed *partial $q^\pi$-realizability*, which bridges the gap between the $q^*$- and $q^\pi$-realizability paradigms. The central contribution of this work is the establishment of computational hardness results for RL under partial $q^\pi$-realizability for both greedy (i.e., argmax) and softmax policy sets. These results extend prior work on computational hardness in $q^*$- and $v^*$-realizable settings [KLL+23], showing that enlarging the policy class beyond the optimal policy $\pi^*$ does not eliminate the fundamental computational challenges.

A limitation of our current setting is that the feature vectors used for policy parameterization in constructing the policy class $\Pi$ (i.e., $\phi'$ and $\theta'$) differ from the realizability parameters (i.e., $\phi$ and $\theta$), which are central to the learning process. Extending our proof techniques to accommodate partial $q^\pi$-realizability with a unified feature vector (i.e., $\phi' = \phi$) remains a significant challenge and is left for future work. It is also worth noting that our NP-hardness results for RL under partial $q^\pi$-realizability should not be seen as purely negative. While they reflect worst-case complexity, they also highlight the need for efficient algorithms under extra assumptions, such as structural constraints on policies or features. For instance, combining the agnostic RL objective [JLR+23] with linear function approximation captures practical settings where $\epsilon$-optimality is defined relative to a policy class. This connection may offer useful directions for developing sample- or computation-efficient methods.

## Acknowledgments

The authors thank Gellért Weisz and Csaba Szepesvári for their helpful discussions throughout this research. Xiaoqi Tan acknowledges support from Alberta Machine Intelligence Institute (Amii), Alberta Major Innovation Fund, and NSERC Discovery Grant RGPIN-2022-03646.

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

## A  Extended Discussion of Related Work

Under the linear function approximation framework, there are two well-defined problem settings in the literature: the $q^*$-realizability and $q^\pi$-realizability settings. Their definitions are provided below.

**Definition A.1** ($q^*$**-Realizable MDP**). *An MDP $M$ is called $q^*$-realizable if there exists $\theta_h \in \mathbb{R}^d$ for any $h \in [H]$ such that for optimal policy $\pi^*$, $q_h^*(s, a) = \langle \phi(s, a), \theta_h \rangle$ holds for all $s \in \mathcal{S}$ and $a \in \mathcal{A}$.*

**Definition A.2** ($q^\pi$**-Realizable MDP**). *An MDP $M$ is called $q^\pi$-realizable if there exists $\theta_h \in \mathbb{R}^d$ for any $h \in [H]$ such that $q_h^\pi(s, a) = \langle \phi(s, a), \theta_h \rangle$ holds for all $\pi, s \in \mathcal{S}$ and $a \in \mathcal{A}$.*

For clarity, in Table 1 we summarize the common assumptions used in both the $q^\pi$-realizability and $q^*$-realizability settings.

| Assumptions | Description |
|---|---|
| Exact realizability ① | $q_h(s, a) = \langle \phi(s, a), \theta_h \rangle + \beta$, where $\beta = 0$ |
| Approximate realizability ② | $q_h(s, a) = \langle \phi(s, a), \theta_h \rangle + \beta$, where $\beta \neq 0$ |
| Access model ③ | Generative model ($\mathcal{M}_g$) or online access ($\mathcal{M}_o$) |
| Minimum suboptimality gap ④ | $\zeta_h(s, a) = V_h^* - q_h^*(s, a) \geq \delta$ |
| Low Variance Condition ⑤ | $E_{s \sim \mu}[|V^\pi(s) - V^*(s)|^2] \leq n_0 (E_{s \sim \mu}[|V^\pi(s) - V^*(s)|])^2$ |
| DSEC Oracle Access ⑥ | Tests if a predictor generalizes across distributions over all $(s, a)$ pairs |
| Action set size ⑦ | For exponential variation ($\exp(d)$) |
| Transition Dynamics ⑧ | For deterministic variation ($\mathcal{P}_{det}$) |
| Bounded feature vector ⑨ | $\|\phi(s, a)\|_2 \leq 1$ for all $(s, a) \in (\mathcal{S} \times \mathcal{A})$ |
| Estimate Oracle Access ⑩ | Computes approximate value function for any state-action pair |
| Spanning feature vectors ⑪ | Matrix $\Phi \in \mathbb{R}^{k \times d}$ spans $\mathbb{R}^d$ with unique rows |
| Feature vector hypercontractivity ⑫ | Anti-concentration of distribution over state-action pairs |

Table 1: List of common assumptions used in related work regarding the $q^*$- and $q^\pi$-realizability settings. Note that here $\beta$ is a linear approximation error; $\mathcal{M}_g$ and $\mathcal{M}_o$ represent the generative model and online access model, respectively. In addition, $\zeta_h(s, a)$ denotes the suboptimality gap for each $(s, a) \in \mathcal{S} \times \mathcal{A}$ at stage $h$. Each of the assumptions comes with a number in front of it, and we use them to indicate the main problem setting of the related work provided in the following tables.

We begin by reviewing results for the $q^*$-realizability setting. Research in this category is generally divided between hardness analyses and upper bound guarantees. Exponential lower bounds under specific conditions are reported by [WAS21], [WWK21], and [DKWY20], as summarized in Table 2. These results rely on assumptions about the structural properties of the MDP (e.g., number of actions, variance constraints, and feature vector structure) and the type of agent-environment interaction (e.g., online, local, or random access). Despite some variations—such as low variance conditions or a minimum suboptimality gap under the online setting—the exponential lower bound remains robust. Nonetheless, polynomial-time algorithms have been proposed under additional assumptions, as detailed by [WWK21], [WR16], [DKWY20], [DLWZ19], and [DLMW20].

There is also a growing body of work focused on the $q^\pi$-realizability setting. For example, [DKWY20] establishes an exponential lower bound in the horizon $H$. However, with additional assumptions—such as bounded feature vectors, access to generative models, estimation oracles, or informative state-action pairs (core set access) as in [LSW20], [YHAY+22], and [WGKS22]—polynomial-time algorithms have been developed. These assumptions are summarized in Table 3.

| Assumptions | Lower bound | Upper bound | Related works |
|---|---|---|---|
| ①, ⑦ ($\exp(d)$) and ③ ($\mathcal{M}_g$) | $\min(e^{\Omega(d)}, \Omega(2^H))$ | LSVI with $\tilde{O}\left(\frac{H^5 d^{H+1}}{\delta^2}\right)$ | [WAS21] |
| ④, ⑤, ③ ($\mathcal{M}_o$), ⑫ | $2^{c \cdot \min(d, H)}$, $c$ constant | poly$\left(\frac{1}{\epsilon}\right)$ queries for $\epsilon$-optimal policy | [WWK21] |
| ⑧ ($\mathcal{P}_{det}$) | - | poly(Eluder Dimension) or poly($\dim_e[Q]$) | [WR16] |
| ④, ⑧ ($\mathcal{P}_{det}$) | - | poly($\dim_e$) with error $\delta = O\left(\frac{\text{gap}}{\dim_e}\right)$ | [DLMW20] |
| ④, ③ ($\mathcal{M}_g$), ② | Exponential | poly($d, H, \frac{1}{\rho}, \log\frac{1}{\delta}$), $\rho$: gap, $\delta$: failure prob. | [DKWY20] |
| ④, ⑤, ⑥ | - | poly$\left(\frac{1}{\epsilon}\right)$ | [DLWZ19] |

Table 2: Sample complexity results for the $q^*$-realizability setting.

| Assumptions | Lower bound | Upper bound | Related works |
|---|---|---|---|
| ②, ③ ($\mathcal{M}_g$) | Exponential in $H$ for $\epsilon = \Omega(\sqrt{\frac{H}{d}})$ | - | [DKWY20] |
| ②, ③ ($\mathcal{M}_g$), ⑫ | - | $\frac{d}{\epsilon^2(1-\gamma)^4}$ with $\epsilon$ accuracy | [LSW20] |
| ②, ③ ($\mathcal{M}_g$), ⑩ | - | $\tilde{O}\left(\frac{d}{\epsilon^2(1-\gamma)^4}\right)$ with $\epsilon$ accuracy | [WGKS22] |

Table 3: The lower and upper bounds in the $q^\pi$-realizability setting under various assumptions. Under the generative model, the statistical hardness is exponential in horizon with approximation error [DKWY20]. Also, in the upper bounds, we can see the dependency on $d$ and $\frac{1}{\epsilon^2}$, where $d$ is the size of the feature vector and $\epsilon$ is an approximation error.

| Main Contribution | Lower Bound | Related Works |
|---|---|---|
| Unique-3-SAT $<_p$ Linear-3-RL | Quasi-polynomial in size of feature vector, $d^{O\left(\frac{\log(d)}{\log\log(d)}\right)}$ | [KLLM22] |
| ($\epsilon$,b)-Gap-3-SAT $<_p$ Linear-3-RL | Exponential, $\exp\left(\tilde{O}\left(\min\left(d^{\frac{1}{4}}, H^{\frac{1}{4}}\right)\right)\right)$ | [KLL+23] |

Table 4: Computational hardness results under the $q^*, v^*$-realizability setting. Note that $A <_p B$ implies a polynomial-time reduction from problem $A$ to problem $B$.

Despite these developments, efficient computation remains a fundamental challenge. Recent studies by [KLLM22] and [KLL+23] emphasize the gap between statistical and computational efficiency under optimal value function realizability, suggesting that NP-hardness may still arise even when sample efficiency is achievable.

Under the $q^*, v^*$-realizability assumption, two important studies investigate computational hardness, summarized in Table 4. In [KLLM22], the authors establish a quasi-polynomial lower bound on the feature dimension $d$, specifically $d^{O\left(\frac{\log(d)}{\log\log(d)}\right)}$, under the randomized exponential time hypothesis (rETH). However, this bound is not tight, as the reduction technique used may leak information about the optimal action after a quasi-polynomial number of interactions, resulting in an imprecise lower bound. In contrast, [KLL+23] presents a stronger exponential lower bound of $\exp(\tilde{O}(\min(d^{1/4}, H^{1/4})))$, where $H$ denotes the horizon and $d$ is the feature dimension. This result demonstrates that the computational hardness of learning near-optimal policies under $q^*$- and $v^*$-realizability is greater than previously established. By reducing from a variant of 3-SAT, the authors derive an exponential hardness result that provides a tighter complexity bound.

Both [KLLM22] and [KLL+23] employ the *Randomized Exponential Time Hypothesis* (rETH) to establish computational lower bounds within the $q^*, v^*$-realizability framework. The core idea of their proofs is to construct an MDP instance from a given Unique-3-SAT instance (a variant of 3-SAT with a unique satisfying assignment), while preserving the structural features relevant to their RL setting. We now describe the general instance design framework used in [KLLM22], as variations of this approach are useful for our reduction in the partial $q^\pi$-realizability setting in Section 3. According to [KLLM22], when designing an MDP instance $M_\varphi$ from a Unique-3-SAT formula $\varphi$, each MDP state represents a candidate assignment, encoded as a tuple of variable values. At each state, the agent is presented with a set of unsatisfied clauses and can choose to flip variables—i.e., to assign them "True" or "False." The agent's objective is to take actions that reduce the Hamming distance to the unique satisfying assignment. The reward structure is randomized in a way that encourages minimizing this distance, thereby aligning the agent's learning objective with the solution to the original SAT instance. The reward is given at the last stage of the MDP or when the satisfying assignment is found, and it relies on the Bernoulli distribution. In this work, due to the structure of the learned policy, a deterministic reduction is also applied to argue the hardness result.

# B  An Example of the MDP Instance $M_\varphi$

In this section, we illustrate the process of designing the MDP instance $\mathcal{M}_\varphi$ through an example based on the $\delta$-MAX-3SAT instance presented in Example 4.1. That example demonstrates how the components of the MDP are constructed from a given $\delta$-MAX-3SAT instance. However, function approximation has not yet been integrated into the framework. As outlined in Sections 4.2.2 and 4.2.3,

the design principles for the PSP and partial realizability vectors have already been established. To improve clarity, we now explicitly construct these parameters for the specific case described in Example 4.1.

Recall that the complexity problem is given by $\varphi = (x_1 \vee \bar{x}_2 \vee x_3) \wedge (\bar{x}_1 \vee x_2 \vee \bar{x}_3)$, which involves three variables and two clauses. Figure 4.2.1 depicts the transformation of this $\delta$-MAX-3SAT instance into an MDP instance. However, the attributes required to define our GLINEAR-2-RL problem instance are not yet fully specified, as the realizability parameters and a well-defined policy set remain to be constructed. As the first step, we now design a greedy policy set for this example using the feature representation $\phi'$.

**PSP Parameters Design.** As discussed in Section 4.2.2, we design the matrix $\Theta \in \mathbb{R}^{d' \times 2^{d'}}$, consisting of $2^{d'}$ vectors, where each $\theta \in \mathbb{R}^{d'}$ represents a greedy policy. In our construction, we set $d' = n$, and specifically, $d' = 3$. Under this setup, the greedy set ensures that the states $\mathcal{S}_h$ at each stage $h$ share the same greedy action, while allowing greedy actions to be selected independently across different stages. In this example, $|\Pi^g| = 8$, meaning that there are eight PSP weight vectors $\theta' \in \mathbb{R}^3$, each corresponding to a unique combination of elements from $\{-1, 1\}^3$.

In the MDP structure depicted in Figure 4.2.1, for the first state, we have:
$$\phi'\big((-1, -1, -1), \text{"True"}\big) = [1, 0, 0],$$
$$\phi'\big((-1, -1, -1), \text{"False"}\big) = [-1, 0, 0].$$

Recall that we always update the $h$-th element of $\phi'(s_h, a_h)$ based on the action $a_h$. Therefore, at each stage $h$, the value of $\phi'(s_h, a_h)$ remains consistent for all $(s_h, a_h) \in \mathcal{S}_h \times \mathcal{A}$, since it depends solely on the stage.

**Partial Realizability Parameters Design.** We begin by demonstrating the representation of $\phi(s, a) \in \mathbb{R}^d$ for all state-action pairs in the MDP shown in Figure 4.2.1, and then illustrate how $\theta_h$ is assigned for any $h \in [H]$ to verify Proposition 4.1. As stated in the proof of Proposition 4.1, we have $d = \ell_1 + \ell_2 + \ell_3$, which implies:
$$d = \binom{2n}{1} + \binom{2n}{2} + \binom{2n}{3} - 2n^2 + n + 1.$$

For $n = 3$, this yields $d = 27$. To facilitate analysis, consider a specific state at the stage $h = 2$, namely $(1, -1, -1)$, which lies along the blue path in Figure 4.2.1. When referring to the values of $\phi$, we focus on $Y_h$ and $b_h$, omitting the constant $\frac{1}{|\mathcal{C}|}$ for simplicity. In this state, the first variable $x_1$ is set to 1, while the remaining variables ($x_2$ and $x_3$) remain undecided. Assume without loss of generality that the greedy policy $\pi \in \Pi^g$ selects the action "True" for all states. From the previous section, this corresponds to $\theta^{\mathsf{T}} = [1, 1, 1]$.

We now examine how the feature vector $\phi((1, -1, -1), \text{"False"})$ is determined. In this example, $b_2$ represents the number of satisfied clauses after setting $x_1 = 1$. Since the first clause of $\varphi$ is satisfied, we have $b_2 = 1$. Following the structure in the proof of Proposition 4.1, we compute $Y_2$. After taking the action "False" in state $(1, -1, -1)$, $x_2$ is set to 0 in the formula $\varphi$. Consequently, the remaining undecided clause is $\mathcal{U}_2 = \{\bar{x}_3\}$. The corresponding entry for $\bar{x}_3$ in $Y_2$ is set to 1 since it appears once, while all other entries in $Y_2$ are set to 0.

Next, we specify the value of $\theta_h$ to ensure partial linear realizability. Given that $x_1$ and $x_2$ have been determined, the component of $\theta_h$ corresponding to $b_2$ is set to 1. For the component corresponding to the clause $\bar{x}_3$, we have $f_2(\theta') = 0$, which maintains consistency with policy $\pi$. Thus, the dot product $\phi((1, -1, -1), \text{"False"}) \cdot \theta_h^{\mathsf{T}}$ evaluates to:
$$\phi\big((1, -1, -1), \text{"False"}\big) \cdot \theta_h^{\mathsf{T}} = \frac{1}{|\mathcal{C}|}\big(b_2 \cdot 1 + 1 \cdot 0\big) = \frac{1}{2}.$$

This matches the value of $q^\pi((1, -1, -1), \text{"False"})$, thereby validating the realizability condition stated in Proposition 4.1.

## C  Remaining Proof of Proposition 4.1

Given a $\delta$-MAX-3SAT instance $\varphi$ with $n$ variables, let $\mathcal{C}_{\text{total}}$ denote the set of all *valid clauses* that can be formed using the given $\delta$-MAX-3SAT variables, excluding trivial clauses—i.e., those of the

form $(x_i \vee \bar{x}_i)$ or $(x_i \vee \bar{x}_i \vee x_j)$, which are always satisfied (evaluating to 1 for any assignment). In particular, we assume w.l.o.g. that the clauses in $\mathcal{C}_{\text{total}}$ are organized in the following order:

$$\mathcal{C}_{\text{total}} = \left\{ \underbrace{c_1^{(1)}, \cdots, c_{\ell_1}^{(1)}}_{\mathcal{C}_{\text{total}}^{(1)}}, \underbrace{c_1^{(2)}, \cdots, c_{\ell_2}^{(2)}}_{\mathcal{C}_{\text{total}}^{(2)}}, \underbrace{c_1^{(3)}, \cdots, c_{\ell_3}^{(3)}}_{\mathcal{C}_{\text{total}}^{(3)}} \right\},$$

where $c_i^{(j)}$ denotes the $i$-th $j$-CNF clause (each clause has exactly $j$ variables) and $\ell_j$ denotes the total number of valid $j$-CNF clauses. Note that $\ell_1 = \binom{2n}{1}$, $\ell_2 = \binom{2n}{2} - n$, and $\ell_3 = \binom{2n}{3} - 2n^2 + 2n$.

**Partial Realizability Feature Vector $\phi$.** Recall that for any state $s_h = (x_1, \cdots, x_{h-1}, -1, \cdots, -1)$ at the beginning of stage $h \in [H]$ (prior to taking action $a_h$), the first $h - 1$ variables have been assigned values of either 0 or 1, based on actions taken before stage $h$. These are referred to as the *assigned variables* $\mathbf{x}_{1:h-1} = (x_1, \cdots, x_{h-1})$, while the remaining $n + 1 - h$ elements, denoted as $\mathbf{x}_{h:n} = (x_h, \cdots, x_n)$, remain unassigned, hence termed *unassigned variables*. For any state-action pair $s_h \in \mathcal{S}_h$ and $a_h \in \mathcal{A}$, let $\mathcal{U}_h$ represent the list of undecided clauses in $\varphi$ after excluding the first $h$ assigned variables (i.e., $x_1$ through $x_h$). Therefore, $\mathcal{U}_0$ includes all clauses initially given by the instance $\varphi$, where $\mathcal{U}_0 = \mathcal{C} \subseteq \mathcal{C}_{\text{total}}$. Upon transitioning to a new state $s_{h+1} \in \mathcal{S}_{h+1}$, the set $\mathcal{U}_h$ is updated, leading to two possible events in $\mathcal{U}_h$. First, since the variable $x_h$ is determined at this stage, certain undecided clauses may become satisfied and are consequently removed from $\mathcal{U}_{h+1}$. Second, some clauses may be modified, reducing in size based on the assigned value of $x_h$. For better illustration of $\mathcal{U}_h$'s construction, let us look at the following example:

$$\varphi : (\bar{x}_1 \vee \bar{x}_2) \wedge (x_1 \vee \bar{x}_4 \vee x_5) \wedge (x_2 \vee \bar{x}_4 \vee x_5) \wedge (x_3 \vee \bar{x}_6 \vee x_7).$$

We have $\mathcal{U}_0 = \{(\bar{x}_1 \vee \bar{x}_2), (x_1 \vee \bar{x}_4 \vee x_5), (x_2 \vee \bar{x}_4 \vee x_5), (x_3 \vee \bar{x}_6 \vee x_7)\}$. At the end of stage $h = 2$, suppose $x_1 = 0$ and $x_2 = 0$. In this case, the first clause $(\bar{x}_1 \vee \bar{x}_2)$ can be decided, while the remaining three clauses stay undecided. However, the undecided clauses $(x_1 \vee \bar{x}_4 \vee x_5)$ and $(x_2 \vee \bar{x}_4 \vee x_5)$ shrink and simplify to $(\bar{x}_4 \vee x_5)$. Thus, at the end of stage $h = 2$, the list of (simplified) undecided clauses in $\varphi$ is $\mathcal{U}_2 = \{(\bar{x}_4 \vee x_5), (\bar{x}_4 \vee x_5), (x_3 \vee \bar{x}_6 \vee x_7)\}$, where the clause $(\bar{x}_4 \vee x_5)$ repeats twice. Using the above concepts, we can now explain how the partial realizability feature vector $\phi$ is constructed. For any $h \in [H]$, let $Y_h$ be a vector of natural numbers, defined as follows:

$$Y_h = \left( y_{h,1}^{(1)}, \ldots, y_{h,\ell_1}^{(1)}, y_{h,1}^{(2)}, \ldots, y_{h,\ell_2}^{(2)}, y_{h,1}^{(3)}, \ldots, y_{h,\ell_3}^{(3)} \right),$$

where $y_{h,i}^{(j)}$ denotes the number of times $c_i^{(j)}$ repeats in $\mathcal{U}_h$. It should be noted that $y_{h,i}^{(j)} = 0$ occurs if the corresponding clause $c_i^{(j)}$ does not appear in the $\delta$-MAX-3SAT instance $\varphi$ or has been decided (either True or False) at the end of stage $h$. For any $s_h \in S_h$ and $a \in A$, we define the partial realizability feature vector $\phi$ as follows:

$$\phi(s_h, a) = \frac{1}{|\mathcal{C}|} \cdot [b_h, Y_h], \qquad \forall h \in [H], \tag{11}$$

where $b_h = |\mathcal{C}_{\text{true}}(s_h)|$, denoting the number of satisfied clauses in $\varphi$ at the end of stage $h$ (i.e., after assigning variable $x_h$). Based on the definition of $\mathcal{U}_h$, we have $b_h \leq |\mathcal{C}| - |\mathcal{U}_h|$, indicating that the number of satisfied clauses is always upper bounded by the total number of clauses that have been decided at the end of stage $h$. We further construct a partially realizable weight vector $\theta_h$ such that the realizability condition stated in Proposition 4.1 is satisfied.

**Partial Realizability Weight Vector $\theta_h$.** We now demonstrate the existence of $\theta_h$ such that the linear realizability condition $q_h^\pi(s_h, a_h) = \langle \phi(s_h, a_h), \theta_h \rangle$ holds. Let $f_h : \mathbb{R}^{d'} \to \{0, 1\}$ be defined as

$$f_h(\theta') = \begin{cases} 0 & \text{if } \theta'_h \leq 0, \\ 1 & \text{otherwise,} \end{cases} \tag{12}$$

where $\theta'_h$ denotes the $h$-th element of $\theta'$. For any state $s_h$ with unassigned elements (i.e., any state with at least one element being -1), we define the virtual *look-ahead* state $\hat{s}_h$ as follows:

$$\hat{s}_h = (x_1, \cdots, x_{h-1}, \hat{x}_h, \cdots, \hat{x}_n), \tag{13}$$

where $\hat{x}_j = f_j(\theta')$ holds for all $j = h, \ldots, n$. Thus, the look-ahead state $\hat{s}_h$ mirrors the current state $s_h$ for the first $h - 1$ elements and matches the final state $s_H$ (i.e., the terminal state at stage $H$ by

following policy $\pi$ in state $s_{h+1}$ onwards) for the last $H - h$ elements (hence, the term "look-ahead"). Given $s_h \in \mathcal{S}_h$ and $a_h \in \mathcal{A}$, we know that the next state is $s_{h+1} = (x_1, \cdots, x_{h-1}, a_h, -1, \cdots, -1)$ and the first $h$ variables (i.e., $\mathbf{x}_{1:h} = (x_1, \cdots, x_h)$) will be assigned.

Let $M_h$ be defined as a vector of 0's and 1's as follows:

$$M_h = \left( m_{h,1}^{(1)}, \cdots, m_{h,\ell_1}^{(1)}, m_{h,1}^{(2)}, \cdots, m_{h,\ell_2}^{(2)}, m_{h,1}^{(3)}, \cdots, m_{h,\ell_3}^{(3)} \right),$$

where $m_{h,i}^{(j)} = 0$ indicates that the clause $c_i^{(j)}$ consists of at least one assigned variable from $\mathbf{x}_{1:h}$. For any clause $c_i^{(j)} \in \mathcal{C}_{\text{total}}^{(j)}$ that depends solely on the unassigned variables $\mathbf{x}_{h+1:n} = (x_{h+1}, \cdots, x_n)$ (i.e. the clause is constructed only based on unassigned variables), let $m_{h,i}^{(j)}$ represent its satisfiability result, assuming that $\mathbf{x}_{h+1:n}$ follows the values of the look-ahead state $\hat{s}_h$ (i.e., $x_j = \hat{x}_j$ for all $j = h + 1, \cdots, H$). Thus, $m_{h,i}^{(j)} = 0$ may occur in any of the following three scenarios: (i) $c_i^{(j)}$ depends only on $\mathbf{x}_{1:h}$ and can therefore be decided, (ii) $c_i^{(j)}$ is a mixture of $\mathbf{x}_{1:h}$ and $\mathbf{x}_{h+1:n}$, or (iii) $c_i^{(j)}$ depends solely on $\mathbf{x}_{h+1:n}$, with a satisfiability result of False based on the look-ahead state. We emphasize that all non-zero elements in $M_h$ (i.e., entries of 1's) are independent of $s_h$ and $a_h$ and are determined solely by $\{f_j(\theta')\}_{\forall j = h+1, \cdots, n}$. Based on the $M_h$ vector, let $\theta_h$ be defined as follows:

$$\theta_h^{\mathsf{T}} = [1, M_h], \qquad \forall h \in [H]. \tag{14}$$

Given the proposed $\phi$ and $\theta_h$, Proposition 4.1 will follow if $q_h^\pi(s_h, a_h) = \frac{1}{|\mathcal{C}|} \cdot (b_h + \langle Y_h, M_h \rangle)$ holds for all $s_h \in \mathcal{S}_h$, $a_h \in \mathcal{A}$, and $\pi \in \Pi^g$.

**Realizability Validation.** Here, we want to verify the linear realizability of our designed MDP that satisfies all the conditions of GLINEAR-2-RL. Given $\phi(s_h, a) = \frac{1}{|\mathcal{C}|} \cdot [b_h, Y_h]$ as demonstrated in Equation (11), we will prove by induction that there exists $\theta_h^{\mathsf{T}} = [1, M_h]$ as described in Equation (14) such that the realizability condition in Proposition 4.1 holds for any $h \in [H]$.

*Base case 1*: We first show that realizability holds for any state $s_h \in \mathcal{S}_h$ at stage $h = H - 1$. Consider a specific state $s_{H-1}$, where the number of satisfied clauses after taking action $a_{H-1}$ is denoted by $b_{H-1}$. If we take any action $a \in \mathcal{A}$ in $s_{H-1}$, the process transitions to the terminal state $s_H$, yielding a reward of $\frac{|\mathcal{C}_{\text{true}}(s_H)|}{|\mathcal{C}|}$. Observe that the reward received after executing action $a_{H-1}$ in $s_{H-1}$ equals $\frac{b_{H-1}}{|\mathcal{C}|}$, i.e., $q_{H-1}^\pi(s_{H-1}, a_{H-1}) = \frac{b_{H-1}}{|\mathcal{C}|}$. After this action, all variables in $\varphi$ become determined, implying that $|\mathcal{U}_{H-1}| = 0$. This implies:

$$
\begin{aligned}
& q_{H-1}^\pi(s_{H-1}, a_{H-1}) \\
&= \langle \phi(s_{H-1}, a_{H-1}), \theta_{H-1} \rangle \\
&= \frac{1}{|\mathcal{C}|} \left( \langle Y_{H-1}, M_{H-1} \rangle + b_{H-1} \right) \\
&= \frac{1}{|\mathcal{C}|} \left( 0 + b_{H-1} \right) \\
&= \frac{b_{H-1}}{|\mathcal{C}|}.
\end{aligned}
\tag{15}
$$

Thus, there exists $\theta_{H-1} = [M_{H-1}, 1]^{\mathsf{T}}$ that the realizability holds.

*Base case 2*: In the first base case, the linear realizability of state $s_{H-1}$ depends solely on the scalar quantity $b_{H-1}$, and we have not yet established how the inner product $\langle Y_h, M_h \rangle$ contributes to the linear realizability condition. To proceed, we now examine realizability for the state $s_{H-2}$. In this state, upon taking action $a_{H-2}$, the system transitions to state $s_{H-1}$, where the agent subsequently follows the greedy policy determined by the look-ahead state $\hat{s}_{H-1}$.

Note that after taking action $a_{H-2}$, the number of satisfied clauses up to stage $H - 2$ is denoted by $b_{H-2}$. The set $\mathcal{U}_{H-2}$ contains the undecided clauses, namely those that do not yet have all their variables determined by assignments $x_1$ through $x_{H-2}$. Moreover, the greedy action selected in state $s_{H-1}$ depends on the elements of $\Theta$. Specifically, by substituting the value of the undecided variable $x_{H-1}$ into $\mathcal{U}_{H-2}$, we can determine all remaining clauses in the $\delta$-MAX-3SAT instance. This substitution is performed according to the greedy action taken in $s_{H-1}$, meaning that the value

of $x_{H-1}$ is determined by the greedy policy and assigned as $f_{H-1}(\theta')$ for each vector $\theta'$ in the matrix $\Theta$. Based on this construction, $\langle Y_{H-2}, M_{H-2}\rangle$ represents the number of clauses that are satisfied from stage $H-1$ onward when following the greedy policy $\pi$. Thus, we have:

$$
\begin{aligned}
q_{H-2}^{\pi}(s_{H-2}, a_{H-2}) &= \langle \phi(s_{H-2}, a_{H-2}), \theta_{H-2}\rangle \\
&= \frac{1}{|\mathcal{C}|}\left(\langle Y_{H-2}, M_{H-2}\rangle + b_{H-2}\right) \\
&= \frac{|\mathcal{C}_{\text{true}}(s_H)|}{|\mathcal{C}|}.
\end{aligned}
\tag{16}
$$

*Induction step*: Given the number of satisfied clauses up to stage $h$ (i.e., $b_h$) and the weight matrix $\Theta$, suppose that the linear realizability condition holds for all state-action pairs $(s_i, a_i) \in \mathcal{S}_i \times \mathcal{A}$ at every stage $i \geq h$ under any greedy policy $\pi \in \Pi^g$, that is, $q_i^{\pi}(s_i, a_i) = \langle \phi(s_i, a_i), \theta_i\rangle$. Our goal is to establish that this linear realizability condition also holds at stage $h-1$ for all state-action pairs and for any greedy policy $\pi \in \Pi^g$. In other words, we aim to prove that:

$$
q_{h-1}^{\pi}(s_{h-1}, a_{h-1}) = \langle \phi(s_{h-1}, a_{h-1}), \theta_{h-1}\rangle \quad \text{for any } (s_{h-1}, a_{h-1}) \in \mathcal{S}_{h-1} \times \mathcal{A}.
$$

Note that $q_{h-1}^{\pi_{h-1}}(s_{h-1}, a_{h-1}) = \frac{|\mathcal{C}_{\text{true}}(s_H)|}{|\mathcal{C}|}$ and $q_h^{\pi_h}(s_h, a_h) = \frac{|\hat{\mathcal{C}}_{\text{true}}(s_H)|}{|\mathcal{C}|}$. Let $z_2 = b_h - b_{h-1}$ denote the difference between the numbers of satisfied clauses until stages $h-1$ and $h$ in $\varphi$. We prove the induction step as follows under policy $\pi \in \Pi^g$:

$$
\begin{aligned}
q_{h-1}^{\pi}(s_{h-1}, a_{h-1}) &= R(s_{h-1}, a_{h-1}) + v_h^{\pi}(s_h) \\
&= q_h^{\pi}(s_h, \pi(s)) = \langle \phi(s_h, \pi(s_h)), \theta_h\rangle \\
&= \frac{1}{|\mathcal{C}|} \cdot \left(\langle Y_h, M_h\rangle + b_h\right) \\
&= \frac{1}{|\mathcal{C}|} \cdot \left(\left(\langle Y_h, M_h\rangle + z_2\right) + (b_h - z_2)\right) \\
&= \frac{1}{|\mathcal{C}|} \cdot \left(\left(\langle Y_h, M_h\rangle + z_2\right) + b_{h-1}\right).
\end{aligned}
\tag{17}
$$

To complete the proof, it suffices to show that $\langle Y_h, M_h\rangle + z_2 = \langle Y_{h-1}, M_{h-1}\rangle$. This is equivalent to demonstrating that $\langle Y_{h-1}, M_{h-1}\rangle - \langle Y_h, M_h\rangle = z_2 = b_h - b_{h-1}$. Since we follow the greedy policy $\pi$ from state $s_h$, the resulting terminal state $s_H$ will be the same whether we begin from $s_{h-1}$ or $s_h$ (after taking action $a_{h-1}$ and subsequently following $\pi$), as both lie along the same trajectory induced by $\pi$. Recall that rewards are only received at the terminal state. Therefore, the cumulative reward from $s_{h-1}$ to $s_H$ must equal that from $s_h$ to $s_H$. This implies $\langle Y_{h-1}, M_{h-1}\rangle + b_{h-1} = b_h + \langle Y_h, M_h\rangle$. Rearranging terms gives the desired equality, thereby completing the inductive step.

Based on the construction of $\phi$ and $\theta_h$, and the fact that the linear realizability condition is satisfied, we complete the proof of Proposition 4.1.

## D  An In-depth Look at Step-2: Algorithmic Connection between $\mathcal{A}_{\text{SAT}}$ and $\mathcal{A}_{\text{RL}}$ under $\Pi^g$

In this section, we detail the second step of the reduction, which establishes the connection between the algorithms $\mathcal{A}_{\text{RL}}$ and $\mathcal{A}_{\text{SAT}}$ in the reduction from $\delta$-MAX-3SAT to the GLINEAR-2-RL problem. Let us first note the following definition regarding the satisfiability ratio:

**Definition D.1** ($\zeta$-Satisfiability). *A* MAX-3SAT *instance is $\zeta$-satisfiable if there is an assignment $x \in \mathcal{X}_{assign}$ such that $\frac{|\mathcal{C}_{true}(x)|}{|\mathcal{C}|} \geq \zeta$.*

We focus on a specific class of $\delta$-MAX-3SAT instances in which there exists an assignment $x \in \mathcal{X}_{\text{assign}}$ that satisfies at least a $(1 - \delta + 2\epsilon)$-fraction of all clauses, where $\epsilon \leq \frac{\delta}{2}$ and $0 < \epsilon, \delta < 1$, where in our setting $\delta = \frac{1}{10}$. This condition aligns with Definition D.1, and throughout this paper, we set $\zeta = 1 - \delta + 2\epsilon$ for our $\delta$-MAX-3SAT instances. In this problem class, it is guaranteed that a minimum $(1 - \delta)$-fraction of clauses can be satisfied, ensuring a valid instance for the reduction

procedure. Given a $\delta$-MAX-3SAT instance that is $(1 - \delta + 2\epsilon)$-satisfiable, we use this decision version of the MAX-3SAT problem to prove the NP-hardness of the GLINEAR-2-RL problem.

To complete the proof of Theorem 3.1, the key idea of our remaining proof is as follows: Given a $(1 - \delta + 2\epsilon)$-satisfiable $\delta$-MAX-3SAT instance $\varphi$, which has a maximum satisfiability ratio of $\frac{|\mathcal{C}^*|}{|\mathcal{C}|}$, we transform it to an instance of GLINEAR-2-RL problem denoted as $M_\varphi$ in polynomial time such that if a random algorithm $\mathcal{A}_{\mathrm{RL}}$ runs on $M_\varphi$, returns $\epsilon$-optimal policy $\pi$ where $\epsilon \leq \frac{\delta}{2}$, then following the policy $\pi$ for setting the value of variables in our $\varphi$, we can satisfy at least $1 - \delta$ fraction of clauses in the given instance $\varphi$.

Consider that $\mathcal{A}_{\mathrm{RL}}$ returns an $\epsilon$-optimal policy $\pi$ in a designed MDP instance $M_\varphi$. It is easy to verify that for any state $s_h \in \mathcal{S}_h$ and for any $h \in [H]$, $v^\pi(s_h) = R(s_{H-1}, \pi(s_{H-1})) = \frac{|\mathcal{C}_{\mathrm{true}}(s_H)|}{|\mathcal{C}|}$. Here, $|\mathcal{C}_{\mathrm{true}}(s_H)|$ denotes the number of clauses satisfied when setting variables in $\delta$-MAX-3SAT problem $\varphi$ following policy $\pi$. Furthermore, we can also denote the value function of the optimal policy $\pi^*$ for any given initial state $s_1$ as follows:

$$v^*(s_1) = \frac{|\mathcal{C}^*|}{|\mathcal{C}|}. \tag{18}$$

Since we know that $\pi^* \in \Pi^g$ based on our constructed $m_\varphi$, we have:

$$\max_{\hat{\pi} \in \Pi^g} v^{\hat{\pi}}(s_1) = \frac{|\mathcal{C}^*|}{|\mathcal{C}|} \geq 1 - \delta + 2\epsilon, \tag{19}$$

where the inequality is due to the fact that the $\delta$-MAX-3SAT instance $\varphi$ is $(1 - \delta + 2\epsilon)$-satisfiable for some $\epsilon \leq \frac{\delta}{2}$. Based on the definition of $\epsilon$-optimality of policy $\pi$ for a given initial state $s_1$, we have

$$|\max_{\hat{\pi} \in \Pi^g} v^{\hat{\pi}}(s_1) - v^\pi(s_1)| \leq \epsilon. \tag{20}$$

Thus, we have

$$|\max_{\hat{\pi} \in \Pi^g} v^{\hat{\pi}}(s_1) - v^\pi(s_1)| = \left| \frac{|\mathcal{C}^*|}{|\mathcal{C}|} - v^\pi(s_1) \right| \leq \epsilon. \tag{21}$$

Note that a $(1 - \delta + 2\epsilon)$-satisfiable instance $\varphi$ is also $(1 - \delta)$-satisfiable. If we want to obtain an output of "Yes" for $\delta$-MAX-3SAT, then we need to have $\frac{|\mathcal{C}_{\mathrm{true}}(s_H)|}{|\mathcal{C}|} \geq 1 - \delta$. This is indeed the case because

$$\frac{|\mathcal{C}_{\mathrm{true}}(s_H)|}{|\mathcal{C}|} \geq \frac{|\mathcal{C}^*|}{|\mathcal{C}|} - \epsilon \geq (1 - \delta + 2\epsilon) - \epsilon \geq 1 - \delta, \tag{22}$$

where the first inequality comes from the definition of $\epsilon$-optimality and the second inequality is due to the fact that $\varphi$ is $(1 - \delta + 2\epsilon)$-satisfiable. As a result, our reduction is completed.

# E    Proof of Theorem 3.2

We begin by presenting an overview of the proof, outlining the two principal components of the reduction. We then formalize each step in detail and conclude by establishing the correctness of Theorem 3.2.

## E.1    An Overview of Our Proof

Building on the framework presented in Section 4.1, we adapt the core proof methodology with minor modifications, which we make explicit in this section. To facilitate our reduction, we introduce a new complexity problem, termed $\delta$-MAX-3SAT($b$), defined as follows:

**Definition E.1** ($\delta$-MAX-3SAT($b$))**.** *Given a MAX-3SAT problem $\varphi$ with $n$ variables, denoted by the set $\mathcal{X} = \{x_1, \ldots, x_n\}$, where each variable $x_i \in \mathcal{X}$ appears in at most $b$ clauses, $\delta$-MAX-3SAT($b$) is defined as the following decision problem: output "Yes" if there exists an assignment $x \in \mathcal{X}_{assign}$ such that $\frac{|\mathcal{C}_{true}(x)|}{|\mathcal{C}|} \geq 1 - \delta$, i.e., at least a $1 - \delta$ fraction of the clauses are satisfied; output "No" otherwise.*

Note that the hardness result of MAX-3SAT mentioned in Lemma 4.1 still holds in the case that each variable appears in at most $b \geq 3$ clauses. Thus, by considering $\delta = 0.1$ and $b = 3$, the $\delta$-MAX-3SAT($b$) problem remains NP-hard under the NP $\neq$ P hypothesis. More specifically, we have the following lemma showing the hardness of $\delta$-MAX-3SAT($b$) under rETH.

**Lemma E.1** (Hardness of $\delta$-MAX-3SAT($b$) under rETH). *Under rETH, there exists constant $b, \delta > 0$ such that no randomized algorithm can solve $\delta$-MAX-3SAT($b$) problem with $n$ variables in time $T = \exp(o(\frac{n}{\text{polylog}(n)}))$ with error probability $\frac{1}{8}$.*[1]

Lemma E.1 establishes that $\delta$-MAX-3SAT($b$) is at least as hard as 3-SAT under rETH. The proof largely follows the arguments presented in [HÅ01, AB09]; however, for completeness, we restate it in Appendix F.

Similar to the hardness proof of GLINEAR-$\kappa$-RL, the key to our proof relies on a reduction process hinges on two critical components:

**Step-1: Polynomial Construction of $M_\varphi$.** We show that it is possible to design an MDP instance $M_\varphi$ (with the given attributes SLINEAR-2-RL) from a given $\delta$-MAX-3SAT($b$) instance $\varphi$ in polynomial time. This requires the polynomial-time construction of the partial realizability features and weight vectors, along with the polynomial-time construction of parametric policies. These steps form the first part of the proof, which is outlined in Section E.2.

**Step-2: Algorithmic Connection between $\mathcal{A}_{\text{SAT}}$ and $\mathcal{A}_{\text{RL}}$.** Recall that the learner interacting with SLINEAR-2-RL using algorithm $\mathcal{A}_{\text{RL}}$, and the algorithm solving $\delta$-MAX-3SAT($b$) is denoted as $\mathcal{A}_{\text{SAT}}$. We show that if $\mathcal{A}_{\text{RL}}$ succeeds on $M_\varphi$ with low error probability, then $\mathcal{A}_{\text{SAT}}$ can be algorithmically derived from $\mathcal{A}_{\text{RL}}$ to solve the $\delta$-MAX-3SAT($b$) instance $\varphi$ with comparable efficiency and low error probability. Specifically, we show that if the $\delta$-MAX-3SAT($b$) instance $\varphi$ is $(1 - \delta)$-satisfiable for some $\epsilon \leq v^* - 3.06 \cdot O\left(\frac{1}{\sqrt{H}}\right) - 0.9$, and algorithm $\mathcal{A}_{\text{RL}}$ returns an $\epsilon$-optimal policy $\pi$ in time $T$ for solving the MDP instance $M_\varphi$ with error probability $\frac{1}{10}$, then following the policy $\pi$ for setting the value of variables in our $\varphi$, we can satisfy at least $1 - \delta$ fraction of clauses in the $\delta$-MAX-3SAT instance $\varphi$ in time $T$ with error probability $\frac{1}{8}$. Finally, we connect our reduction process to rETH by given Lemma E.1, which leads to our Theorem 3.2.

Both components are essential for demonstrating the hardness of SLINEAR-2-RL. Compared to the proof of Theorem 3.1, the key modification here in this section, which distinguishes Theorem 3.1 from Theorem 3.2, arises from the inclusion of *stochastic policies*. This stochastic framework allows us to refine our results through the lens of randomized reduction, and further investigating our hardness result under rETH, enabling a more precise characterization of computational hardness.

## E.2  An In-depth Look at Step-1: Polynomial Construction of $M_\varphi$

Following the framework established in Section 4.1, we construct a polynomial-time reduction from a $\delta$-MAX-3SAT($b$) instance $\varphi$ (as defined in Definition E.1) to an MDP instance that captures the structure of the SLINEAR-2-RL problem. The instance $\varphi$ consists of $n$ Boolean variables ordered as $(x_1, \ldots, x_n)$, with each variable appearing in at most $b$ clauses. Since the core MDP components—states, actions, transitions, and rewards—remain identical to those in the GLINEAR-2-RL framework, we focus exclusively on the modifications made to the approximation-related components. In particular, we describe the polynomial-time construction of the PSP vector and the partial realizability vectors, both of which are specifically adapted to the SLINEAR-2-RL problem.

**Design of Partial Realizability Vectors and Realizability Validation.** We retain the structure of $\phi$ as described in Section 4.2.3, while modifying the structure of $\theta$ to accommodate the softmax policy set. In particular, adjustments are necessary for the virtual look-ahead state $\hat{s}_h$, defined in Equation (13). Recall that in $\hat{s}_h$, the variables $x_1$ through $x_{h-1}$ are determined by the actions taken in the preceding stages, whereas the remaining variables, $\hat{x}_h$ through $\hat{x}_n$, must be selected according to the policy $\pi$.

In Section 4.2.3, the policy $\pi$ was assumed to belong to the set of deterministic greedy policies $\Pi^g$. For any $\pi \in \Pi^g$, the nonzero entries in the vector $M_h$—a key component of $\theta_h^{\mathsf{T}}$—are determined by

---

[1]Note that here, we use a looser bound error probability bound in comparison with the bound in rETH (Definition 2.1). This error probability match the error probability of $\mathcal{A}_{\text{SAT}}$ in our reduction in Section E.3.

tracing the variable assignments along the look-ahead path using the functions $\{f_j(\theta')\}_{j=h+1}^n$, which characterize the behavior of the greedy policy in undecided states.

To generalize this construction to the softmax policy setting, we require a function that similarly governs action selection in undecided states, consistent with the probabilistic nature of softmax-based decisions. Crucially, we show that by suitably modifying the parameters $\theta$, the principle of value function decomposition can still be preserved under this new framework. This leads to the following proposition, which formalizes the linear realizability of value functions in the context of softmax policies:

**Proposition E.1.** *Given a softmax policy set $\Pi^{sm}$, for any state $s_h \in \mathcal{S}_h$, $a_h \in \mathcal{A}$, and $\pi \in \Pi^{sm}$, there exist $\phi(s_h, a) \in \mathbb{R}^d$ and $\theta_h \in \mathbb{R}^d$ such that for any stage $h \in [H]$:*

$$q_h^\pi(s_h, a_h) = \langle \phi(s_h, a_h), \theta_h \rangle. \tag{23}$$

*Proof.* As we stated at the end of Section 4.2.3, the main idea of design is to guarantee that the feature vector $\phi(s_h, a_h)$ depends on $s_h$ and $a_h$, while the weight vector $\theta_h$ depends only on the roll-out of the softmax policy beyond stage $h$, thereby enabling the linear realizability condition $q_h^\pi(s_h, a_h) = \langle \phi(s_h, a_h), \theta_h \rangle$. The approach that we use here again is to decompose the value of each state-action pair into two terms:

$$q_h^\pi(s_h, a_h) = \frac{1}{|\mathcal{C}|} \cdot (b_h + \langle Y_h, M_h \rangle),$$

where the first term, $b_h$, represents the number of satisfied clauses up to stage $h$, and the second term, $\langle Y_h, M_h \rangle$, indicates the number of satisfied clauses when following the softmax policy from stage $h + 1$ to the final stage $H$. Compared to the proof performed under $\Pi^g$, the changes that must be applied in the case of a given softmax policy $\pi \in \Pi^{sm}$ only affect the term $M_h$.

We now proceed to prove linear realizability by appropriately characterizing the structure of $M_h$ under the softmax policy set $\Pi^{sm}$. The main technical challenge in transitioning to $\Pi^{sm}$ lies in constructing a function—analogous to $f_h(\theta')$ from Section 4.2.3, embedded within $\theta_h^\intercal$—that accurately captures the action choices made by softmax policies in undecided states. For ease of exposition, we refer to this function as the *policy follower*.

Our approach begins by analyzing the structure of the action-value function $q_h^\pi(s_h, a_h)$, which provides the foundation for defining the appropriate policy follower under stochastic policies. For any $\pi \in \Pi^{sm}$, the action at each stage is drawn according to the probability distribution $\pi(a_h \mid s_h)$, inducing stochastic trajectories in the MDP $M_\varphi$. Consequently, the value $q_h^\pi(s_h, a_h)$ is expressed as the expected cumulative reward over all possible continuations from $(s_h, a_h)$, that is:

$$q_h^\pi(s_h, a_h) = \mathbb{E}_\pi \left[ R(s_H) \mid s = s_h, a = a_h \right]. \tag{24}$$

The expectation in $q_h^\pi(s_h, a_h)$ accounts for the stochasticity introduced by softmax action selections along the trajectory. For any policy $\pi \in \Pi^{sm}$, when applied to states in stage $h$ (i.e., $s_h \in \mathcal{S}_h$), the resulting action distributions induce a set of possible paths to the terminal state, each corresponding to a different sequence of stochastic decisions. Specifically, this gives rise to $2^{H-h-1}$ distinct transition paths from $(s_h, a_h)$ to the terminal state $s_H$.

Let $\mathcal{T}_h^\pi(s_h, a_h)$ denote the set of all such transition paths generated by taking action $a_h$ in state $s_h$ and subsequently following policy $\pi \in \Pi^{sm}$. Each path $\tau \in \mathcal{T}_h^\pi(s_h, a_h)$ corresponds to a sequence of state-action pairs beginning at $(s_h, a_h)$ and terminating at $s_H$, i.e., $\tau = (s_h, a_h, \ldots, s_H)$. When the context is clear, we will abbreviate this set as $\mathcal{T}_h^\pi$ for notational convenience.

For any $\tau \in \mathcal{T}_h^\pi$, let $P^{(\tau)}$ denote the probability of the path $\tau$ sampled under the softmax policy $\pi$. Let $R^{(\tau)}(s_H)$ denote the reward obtained at the terminal state following the path $\tau$. Clearly, the above expectation over the terminal reward can be written as

$$q_h^\pi(s_h, a_h) = \sum_{\tau \in \mathcal{T}_h^\pi} P^{(\tau)} \cdot R^{(\tau)}(s_H), \tag{25}$$

To ensure the linear realizability condition for $q^\pi(s_h, a_h)$ for any policy $\pi$, we require $q_h^\pi(s_h, a_h) = \langle \phi(s_h, a_h), \theta_h \rangle$. Let $\theta_h^{(\tau)}$ represent a vector obtained by incorporating the policy induced by the

trajectory $\tau$ into the $M_h$ component of $\theta_h$. Thus, we have $\theta_h^{(\tau)} = [1, M_h^{(\tau)}]^\intercal$, where $M_h^{(\tau)}$ denotes the vector $M_h$ when we follow the trajectory $\tau$ for setting values of $M_h$ elements. Suppose that action value function is realizable:

$$
\begin{aligned}
q_h^\pi(s_h, a_h) &= \langle \phi(s_h, a_h), \theta_h \rangle \\
&= \sum_{\tau \in \mathcal{T}_h^\pi} P^{(\tau)} \cdot \langle \phi(s_h, a_h), \theta_h^{(\tau)} \rangle \\
&= \Big\langle \phi(s_h, a_h), \sum_{\tau \in \mathcal{T}_h^\pi} P^{(\tau)} \cdot \theta_h^{(\tau)} \Big\rangle \\
&= \Big\langle \phi(s_h, a_h), \sum_{\tau \in \mathcal{T}_h^\pi} P^{(\tau)} \cdot [1, M_h^{(\tau)}]^\intercal \Big\rangle .
\end{aligned}
\tag{26}
$$

Based on Equation (26), if we design $\theta_h$ by

$$
\theta_h = \sum_{\tau \in \mathcal{T}_h^\pi} P^{(\tau)} \cdot [1, M_h^{(\tau)}]^\intercal, \qquad \forall h \in [H],
\tag{27}
$$

then the linear realizability condition in Equation (23) holds. Note that $P^{(\tau)}$ is determined in $O(H)$ for any policy $\pi \in \Pi^{sm}$, stage $h \in [H]$ and trajectory $\tau$. As a result, each non-zero element of $M_h$ (denoted by $m_{h,i}^{(j)}$ in Section 4.2.3) is a weighted sum of clauses' assignments under different possible paths $\tau \in \mathcal{T}_h^\pi(s_h, a_h, \pi)$ scaled by $P^{(\tau)}$. We thus complete the proof of Proposition E.1. $\qquad \square$

As previously discussed, it is essential to ensure that the reduction from the $\delta$-Max-3SAT instance to the sLinear-2-RL problem can be carried out in polynomial time. From earlier constructions of the partial realizability and PSP feature vectors, we know that both can be computed in $O(n^3)$ time. The remaining component to analyze is the parameter vector $\theta_h \in \mathbb{R}^d$. Each entry of $\theta_h$—which corresponds to elements in $M_h$—can be efficiently computed via a dynamic programming procedure in $O(H)$ time. Given that the dimension $d$ satisfies $d = O(n^3)$, the overall time required to construct $\theta_h$ is $O(n^3 \cdot H) = O(n^4)$, which remains polynomial in the input size $n$. This confirms that the full reduction to the sLinear-2-RL instance is computationally efficient.

### E.3 An In-depth Look at Step-2: Algorithmic connection between $\mathcal{A}_{\text{SAT}}$ and $\mathcal{A}_{\text{RL}}$ under $\Pi^{sm}$

As we mentioned in Section E.1, the second component of the reduction involves establishing a correct algorithmic connection. To complete the proof of Theorem 3.2, it remains to demonstrate how the $\mathcal{A}_{\text{RL}}$ algorithm can be leveraged to construct the $\mathcal{A}_{\text{SAT}}$ method such that solving the sLinear-2-RL instance efficiently yields an efficient solution for the $\delta$-Max-3SAT$(b)$ instance. Our key idea is as follows: Given a $(1 - \delta)$-satisfiable $\delta$-Max-3SAT$(b)$ instance $\varphi$, which has a maximum satisfiability ratio of $\frac{|\mathcal{C}^*|}{|\mathcal{C}|}$. We transform it to an instance of sLinear-2-RL denoted as $M_\varphi$ (the constructed MDP instance) in polynomial time such that if a randomized algorithm $\mathcal{A}_{\text{RL}}$ runs on $M_\varphi$ returns $\epsilon$-optimal policy $\pi$ with error probability $\frac{1}{10}$ in time $T$, then following the policy $\pi$, if we use $\mathcal{A}_{\text{RL}}$ algorithm as a module in designing randomized algorithm $\mathcal{A}_{\text{SAT}}$ for setting the value of variables in our $\varphi$, we can satisfy at least $1 - \delta$ fraction of clauses in time $T$ with error probability $\frac{1}{8}$.

Here, since $\{\pi^*\} \subsetneq \Pi^{sm}$, we only need to establish the reduction based on the best stochastic policy[2] within the set $\Pi^{sm}$. Recall that $R(s_H) = \frac{|\mathcal{C}_{\text{true}}(s_H)|}{|\mathcal{C}|}$ denotes the reward obtained at the terminal state $s_H$. Note that $R(s_H)$ is a random variable due to the stochastic nature of the softmax policy set. We can now state the $\epsilon$-optimality condition as given in Equation (28):

$$
\arg \max_{\hat{\pi} \in \Pi^{sm}} v^{\hat{\pi}} - v^\pi = \frac{|\mathcal{C}^*|}{|\mathcal{C}|} - \mathbb{E}[R(s_H)] \leq \epsilon.
\tag{28}
$$

---

[2]More formally, with high probability, the policy class $\Pi^{sm}$ contains stochastic policies whose value functions closely approximate the optimal value function $v^*$. This is because $\Pi^{sm}$ is constructed from the parameter matrix $\Theta \in \mathbb{R}^{d' \times \infty}$ and encompasses an infinite collection of softmax policies. As a result, the probability that $\Pi^{sm}$ includes a near-optimal policy converges to 1.

By rearranging Equation (28), we have the lower bound on the expected value of the observed sampled rewards.

$$\mathbb{E}[R(s_H)] \geq \frac{|\mathcal{C}^*|}{|\mathcal{C}|} - \epsilon. \tag{29}$$

Note that given an $\epsilon$-optimal policy $\pi \in \Pi^{sm}$ in the initial state $s_1$, we want to ensure that by following the stochastic policy $\pi$ for setting the value of the $\delta$-MAX-3SAT($b$) instance, with high probability, we can satisfy at least a $1 - \delta$ fraction of the clauses. Therefore, we want to ensure that $R(s_H) \geq 1 - \delta$ holds with high probability. To achieve this, we utilize the McDiarmid inequality, which establishes a probabilistic connection between the random variable and its expectation under specific conditions.

**Lemma E.2** (McDiarmid's inequality). *Let $X_1, \ldots, X_n$ be independent random variables, where each $X_i$ takes values in a set $\mathcal{X}_i$. Let $\mathcal{F} : \mathcal{X}_1 \times \cdots \times \mathcal{X}_n \to \mathbb{R}$ be a function satisfying the **bounded differences property**: for every $i = 1, \ldots, n$, if two inputs $(x_1, \ldots, x_n)$ and $(x'_1, \ldots, x'_n)$ differ only in the $i$-th coordinate (i.e., $x_j = x'_j$ for all $j \neq i$), then*

$$|\mathcal{F}(x_1, \ldots, x_n) - \mathcal{F}(x'_1, \ldots, x'_n)| \leq r_i,$$

*where $r_i$ is a constant for each $i$. For any $t > 0$, the deviation of $f$ from its expectation satisfies:*

$$\Pr\left(\mathcal{F}(X_1, \ldots, X_n) - \mathbb{E}[\mathcal{F}(X_1, \ldots, X_n)] \geq t\right) \leq \exp\left(\frac{-2t^2}{\sum_{i=1}^n r_i^2}\right), \tag{30}$$

$$\Pr\left(\mathcal{F}(X_1, \ldots, X_n) - \mathbb{E}[\mathcal{F}(X_1, \ldots, X_n)] \leq -t\right) \leq \exp\left(\frac{-2t^2}{\sum_{i=1}^n r_i^2}\right). \tag{31}$$

Based on the concentration inequality stated in Lemma E.2, consider a function $\mathcal{F}$ comprising $n$ inputs (independent random variables $X_1, X_2, \ldots, X_n$). If altering any single variable $X_i$ within its domain $\mathcal{X}_i$ (while keeping all other variables fixed) results in a bounded change $r_i$ in the value of $\mathcal{F}$, then $\mathcal{F}$ satisfies the *bounded differences property*. For such a function $\mathcal{F}$ satisfying these conditions, the concentration inequality in Equation (30) and Equation (31) holds.

Based on Lemma E.2, we aim to derive a concentration bound for our random variable $R(s_H)$, which is bounded between 0 and 1. Our reward function $R(s_H)$ also satisfies the *bounded difference property* as outlined in Lemma E.2, since changing any action through the trajectory from $s_1$ to $s_H$ only induces a bounded difference in the value of $R(s_H)$. In our $\delta$-MAX-3SAT($b$) instance, since each variable appears in at most $b$ clauses, we have $r_i = \frac{b}{|\mathcal{C}|}$ for any $i \in [n]$, leading to

$$\Pr\left(R(s_H) - \mathbb{E}[R(s_H)] \leq -t\right) \leq \exp\left(\frac{-2t^2 \cdot |\mathcal{C}|^2}{H \cdot b^2}\right). \tag{32}$$

Let us consider $\mathfrak{p}_0 = \exp\left(\frac{-2t^2 \cdot |\mathcal{C}|^2}{H \cdot b^2}\right)$, then $t = \frac{b}{|\mathcal{C}|}\sqrt{\frac{H \ln\left(\frac{1}{\mathfrak{p}_0}\right)}{2}}$. We can rearrange the terms and obtain the complement in Equation (32) as follows:

$$\Pr\left(R(s_H) \geq \mathbb{E}[R(s_H)] - \frac{b}{|\mathcal{C}|}\sqrt{\frac{H \ln\left(\frac{1}{\mathfrak{p}_0}\right)}{2}}\right) \geq 1 - \mathfrak{p}_0. \tag{33}$$

We can now substitute the lower bound on $\mathbb{E}[R(s_H)]$ from Equation (29), leading to the following inequality:

$$R(s_H) \geq \frac{|\mathcal{C}^*|}{|\mathcal{C}|} - \epsilon - \frac{b}{|\mathcal{C}|}\sqrt{\frac{H \ln\left(\frac{1}{\mathfrak{p}_0}\right)}{2}}.$$

Recall that we want to ensure

$$\Pr\left(R(s_H) \geq 1 - \delta\right) \geq 1 - \mathfrak{p}_0$$

for some small probability $\mathfrak{p}_0$. Therefore, based on Equation (33), we need to ensure that the following condition holds:

$$\frac{|\mathcal{C}^*|}{|\mathcal{C}|} - \epsilon - \frac{b}{|\mathcal{C}|}\sqrt{\frac{H \ln\left(\frac{1}{\mathfrak{p}_0}\right)}{2}} \geq 1 - \delta.$$

By rearranging the inequality, we obtain the following condition that must hold for the given $\epsilon$:

$$\epsilon \leq \frac{\mathcal{C}^*}{\mathcal{C}} + \delta - \frac{b}{|\mathcal{C}|}\sqrt{\frac{H\ln(\frac{1}{\mathfrak{p}_0})}{2}} - 1. \tag{34}$$

In Equation (34), we have the following negative term:

$$-\frac{b}{|\mathcal{C}|}\sqrt{\frac{H\ln\left(\frac{1}{\mathfrak{p}_0}\right)}{2}}. \tag{35}$$

To achieve a small error probability $\mathfrak{p}_0$, we can control this term by increasing $|\mathcal{C}|$. Additionally, note that $b \geq 3$ (we choose $b = 3$) and $\delta = 0.1$. Here, we fix error probability $\mathfrak{p}_0 = \frac{1}{8}$. Furtheromore, note that here $n = d'$, $|\mathcal{C}| = O(n) = O(H)$, and $\frac{|\mathcal{C}^*|}{|\mathcal{C}|} = v^*$. As a result, we have that:

$$\epsilon \leq v^* - 3.06 \cdot O\left(\frac{1}{\sqrt{H}}\right) - 0.9. \tag{36}$$

To guarantee that the right hand-side of the above inequality stays positive, we need to have $v^* - 3.06 \cdot O\left(\frac{1}{\sqrt{H}}\right) - 0.9 > 0$, hence the condition $H = \Omega(\frac{1}{(v^*-0.9)^2})$. Based on the mentioned conditions, we have the following probability inequality:

$$\Pr\left(R(s_H) \geq 1 - \delta\right) \geq 1 - \mathfrak{p}_0,$$

where $\mathfrak{p}_0 = \frac{1}{8}$ represents a small probability of failure.

Recall that the goal is to use the randomized algorithm $\mathcal{A}_{\text{RL}}$ as a subroutine within $\mathcal{A}_{\text{SAT}}$, such that, with high probability, at least a $1-\delta$ fraction of the clauses are satisfied. Since the reduction is randomized, we must ensure that the overall success probability—which includes both the correctness of the polynomial-time transformation and the success of solving the $\delta$-MAX-3SAT$(b)$ instance—exceeds $\frac{2}{3}$ (see Chapter 7 of [AB09]). As all components of the instance transformation are deterministic, the success probability of the randomized reduction depends on two factors: the probability that $\mathcal{A}_{\text{RL}}$ returns an $\epsilon$-optimal policy, which is $\frac{9}{10}$, and the probability that $\mathcal{A}_{\text{SAT}}$ succeeds on $\delta$-MAX-3SAT$(b)$ when employing $\mathcal{A}_{\text{RL}}$, which is $1 - \mathfrak{p}_0$. Combining these, the total success probability is $\frac{9}{10} \cdot (1 - \mathfrak{p}_0) > \frac{2}{3}$. This completes the reduction.

## F Hardness Result about $\delta$-MAX-3SAT$(b)$

To establish the hardness result for $\delta$-MAX-3SAT$(b)$ under rETH, as stated in Lemma E.1, we first introduce a complexity problem called $(b,\epsilon)$-GAP-3-SAT, as defined by [KLL+23], which is given as follows:

**Definition F.1** (($(b,\epsilon)$-GAP-3-SAT [KLL+23])**.** *Given a 3-CNF formula $\varphi$ with $v$ variables and $O(v)$ clauses, the following conditions hold:*

- *Each variable appears in at most $b$ clauses.*

- *Either $\varphi$ is satisfiable, or any assignment leaves at least an $\epsilon$-fraction of clauses unsatisfied.*

Based on the hardness result for solving $(b,\epsilon)$-GAP-3-SAT in [KLL+23], we restate the following lemma.

**Lemma F.1** (Hardness of $(b,\epsilon)$-GAP-3-SAT [KLL+23])**.** *Under rETH, there exist constants $b, \epsilon, c > 0$ such that no randomized algorithm can solve $(b,\epsilon)$-GAP-3-SAT with $v$ variables in time $\exp(cv/polylog(v))$ with error probability $1/8$.*

To complete the proof of Lemma E.1, it suffices to reduce $(b,\epsilon)$-GAP-3-SAT to $\delta$-MAX-3SAT$(b)$. For this, we assume very small constants $\epsilon$ and $\delta$ to ensure that $(b,\epsilon)$-GAP-3-SAT satisfies the conditions of Lemma F.1, while preserving the NP-hardness of $\delta$-MAX-3SAT$(b)$. Without loss of generality, we further assume that $\delta > \epsilon$.

Given an instance $\varphi_1$ of $(b,\epsilon)$-GAP-3-SAT with $v$ variables and $|\mathcal{C}|$ clauses, we proceed with the reduction as follows: we copy $\varphi_1$ and create an instance $\varphi_2$ for $\delta$-MAX-3SAT$(b)$. Let the algorithm

that interacts with the $(b,\epsilon)$-GAP-3-SAT instance be denoted as $\mathcal{A}_{\text{gap}}$, and the algorithm that interacts with the $\delta$-MAX-3SAT$(b)$ instance as $\mathcal{A}_{\text{max}}$. If the following conditions hold, we have a successful reduction:

- If $\mathcal{A}_{\text{max}}$ solves $\varphi_2$ and returns an assignment $x \in \mathcal{X}_{\text{assign}}$ such that $\frac{|\mathcal{C}_{\text{true}}(x)|}{|\mathcal{C}|} \geq 1 - \delta$, then following $x$, we satisfy all clauses of $\varphi_1$ and return "Yes".

- If $\mathcal{A}_{\text{max}}$ cannot solve $\varphi_2$ and returns an assignment $x \in \mathcal{X}_{\text{assign}}$ such that $\frac{|\mathcal{C}_{\text{true}}(x)|}{|\mathcal{C}|} < \delta$, then following $x$, we leave at least an $\epsilon$-fraction of clauses unsatisfied in $\varphi_1$ and return "No".

For the first argument, if $\mathcal{A}_{\text{max}}$ returns an assignment $x$ such that $\frac{|\mathcal{C}_{\text{true}}(x)|}{|\mathcal{C}|} \geq 1 - \delta$, then since $\delta > \epsilon$, it implies that all clauses in $\varphi_1$ are satisfied. For the second argument, if the returned assignment $x$ satisfies $\frac{|\mathcal{C}_{\text{true}}(x)|}{|\mathcal{C}|} < 1 - \delta$, then we cannot satisfy all clauses in $\varphi_2$, and at least an $\epsilon$-fraction of the clauses remain unsatisfied. The instance transformation occurs in polynomial time, specifically $O(v)$, and the appropriate algorithmic connection is established. Consequently, we conclude that $\delta$-MAX-3SAT$(b)$ is as hard as $(b,\epsilon)$-GAP-3-SAT, and the hardness result from Lemma F.1 also holds for $\delta$-MAX-3SAT$(b)$ under rETH.

