# OpenReview forum: "Computational Hardness of Reinforcement Learning with Partial $q^{\pi}$-Realizability"
_NeurIPS.cc/2025/Conference — NeurIPS 2025 poster_

### Official Review · Reviewer_v2tA · 2025-06-02

**Clarity:** 4
**Significance:** 4
**Originality:** 4
**Rating:** 5
**Confidence:** 4

**Summary:**

This paper explores the computational complexity of reinforcement learning algorithms under the partial q^pi-realizability setting, a novel framework introduced by the authors.

They first establish NP-hardness when considering a class of greedy policies. Furthermore, under the Randomized Exponential Time Hypothesis, they show that learning with softmax policy classes has exponential computational complexity in the dimension of the feature vector.

The proposed setting and results help bridge the gap between the linear q* and linear q^pi assumptions, offering a more complete understanding of the landscape.

A key technical contribution is the reduction from two well-known complexity problems:  delta-MAX-3SAT and delta-MAX-3SAT(b).

**Questions:**

The proposed partial q^pi setting is quite interesting. Could you provide some natural or practical examples that illustrate this setting?

The paper mentions that related works use reductions to establish hardness results. Could you elaborate on how the reduction used in this paper differs from those in prior work?

**Ethical Concerns:**

["NO or VERY MINOR ethics concerns only"]

**Final Justification:**

After carefully reading the other reviews and the subsequent discussions, I am inclined to maintain my accept recommendation.

I believe this paper will be very solid after revisions based on the feedback provided.

**Quality:**

4

**Strengths And Weaknesses:**

This paper is well motivated, as it explores the intermediate regime between two fundamental settings in reinforcement learning theory: linear q* and linear q^pi.

The discussion of related work is thorough and informative, providing valuable context for the reader.

The assumption that the agent has access to a generative model strengthens the significance of the hardness results presented.

Previous work has shown quasi-polynomial and even exponential lower bounds under the q*- and v* -realizability assumptions. On the other hand, under the stronger assumption of linear q^pi-realizability, efficient algorithms exist when using a generative model. This naturally raises the question: Can the positive results be generalized, or do hardness results still hold under more relaxed assumptions?

This paper provides a compelling answer, demonstrating that computational hardness persists even when the policy class extends beyond the optimal policy pi*, that is, beyond the linear q* setting.

The two main results (Theorems 3.1 and 3.2) are both significant. The core technical contribution lies in clever reductions from classical complexity problems, which underpin the hardness proofs.

Although the proposed partial linear q^pi setting is novel, it appears to lack natural motivating examples.

---

> ### Author Rebuttal · Authors · 2025-07-31
>
> We sincerely thank the reviewer for their positive evaluation and thoughtful feedback. We are especially grateful for the encouraging remarks regarding our motivation, technical contributions, and the significance of the results. Below, we respond to the reviewer’s two follow-up questions concerning (i) natural motivating examples for the proposed partial linear $q^{\pi}$ setting, and (ii) how our reduction approach differs from those in prior hardness results.
>
> ## Answer to Question 1 (Practical Examples)
>
> That is an excellent question, as addressing it can provide insights into both the design of theoretically motivated algorithms and the analysis of current real-world applications from the perspective of partial $q^{\pi}$-realizability. We offer two perspectives for identifying practical examples: one directly motivated by our problem setting, and another arising from an extension of our framework to a broader setting known as Agnostic RL ([1]).
>
> A directly related example, fully aligned with our setting, is that of autonomous driving. In this case, the RL agent (i.e., the automated vehicle) must choose among a set of possible policies, which can be interpreted as feasible trajectories. These policies might be generated by a neural network trained via supervised learning or provided by human experts. Consequently, the policy set is constructed independently of the learning parameters. For instance, one could associate the policy representation parameters with elements such as fuel level, engine power, and the primary navigation system settings. These parameters correspond to our first feature vector, $\phi'$, which governs the policy construction in the partial $q^{\pi}$-realizability setting. In addition to this, another feature vector is used to represent state-action pairs, capturing aspects such as localization and other factors directly tied to the learning process for identifying near-optimal policies (i.e., trajectories).
>
> On the other hand, there is a close connection between partial $q^{\pi}$-realizability and the well-known Agnostic RL framework, where the goal is to find the best policy with respect to a given policy class, even if the optimal policy is not included in that class. Although the Agnostic RL framework is traditionally defined for tabular settings, the conceptual similarity to our problem formulation becomes more apparent as we explore extensions to our model. A key insight here is that, in many real-world scenarios, one might only have access to a finite policy set, and the true optimal policy may not be included due to the complexity of the underlying value function. Returning to the autonomous driving example, this would correspond to a situation where the best possible trajectory (in terms of reward) is not included in the available set due to the limitations of model expressivity or safety constraints.
>
> Identifying such real-world applications---and understanding their connections to partial $q^{\pi}$-realizability---is one of our core goals, as it not only validates the theoretical framework but also offers potential pathways for practical algorithm development.
>
> ## Answer to Question 2 (Comparison of Our Reduction Approach with Prior Work)
> The reduction process typically involves two main steps: (1) designing a polynomial-time reduction from a known complexity problem to an instance of the target problem, and (2) establishing a rigorous algorithmic connection between solutions to the original complexity problem and the constructed instance.
>
> Our approach significantly differs from prior work in the first step, which is arguably the most critical part of the reduction. Specifically, we base our reductions on carefully crafted hardness problems of our own design---namely, $\delta\text{MAX-3SAT}$ and $\delta\text{MAX-3SAT}(b)$. These problems are NP-optimization problems, and designing reductions from them requires more intricate polynomial transformations than those used in prior works such as [2] and [3], where such NP-optimization variants were not employed.
>
> From a high-level perspective, our reduction is also distinct in its ability to handle stochastic policy sets. This enables us to develop a randomized reduction under the softmax policy class, which further differentiates our work in the second step of the reduction, where the connection between solution spaces is established to complete the hardness proof.
>
> **--- [References]---**
>
> [1] Z. Jia, G. Li, A. Rakhlin, A. Sekhari, and N. Srebro. When is agnostic reinforcement learning statistically tractable?, 2023.
>
> [2] D. Kane, S. Liu, S. Lovett, and G. Mahajan. Computational-statistical gaps in reinforcement learning, 2022.
>
> [3] D. Kane, S. Liu, S. Lovett, G. Mahajan, C. Szepesv´ari, and G. Weisz. Exponential hardness of reinforcement learning with linear function approximation, 2023.

---

> > ### Comment · Reviewer_v2tA · 2025-08-04
> >
> > I appreciate the authors for addressing my questions.
> >
> > After carefully reading the other reviews and the subsequent discussions, I am inclined to maintain my accept recommendation.
> >
> > Thank you for the excellent work. I believe this paper will be very solid after revisions based on the feedback provided.

---

### Official Review · Reviewer_SdE3 · 2025-06-26

**Clarity:** 4
**Significance:** 3
**Originality:** 3
**Rating:** 5
**Confidence:** 3

**Summary:**

This paper presents a new intermediate computational problem called _partial $q^{\pi}$-realizability_, along with new complexity analysis for this setting. Prior work has examined two end points related to this new setting termed $q*$-realizability (in which $q^*$ is representable using a linear combination invoking features, $\phi$) and $q^\pi$-realizability (in which $q^\pi$ is representable for all policies). Here, the problem of partial $q^\pi$-realizability_ is defined _relative_ to a choice of policy class, $\Pi$, which allows the problem to soften its requirements: only the $q$ functions of the policies contained in the set must be representable. Then, the second contribution of the work is new analysis of this partial realizability problem. The analysis comes in two Theorems which differ in which policy class the result centers around. The first (Theorem 3.1), makes use of the class of deterministic greed policies, $\Pi^g$, while the second (Theorem 3.2) uses softmax policies; in both cases, the corresponding versions of the problem are proven to be hard. The rest of the paper is dedicated to an overview of the proof technique, involving details about the reductions used and examples.

**Questions:**

I only have a few high-level suggestions:
- You might consider restructuring Sections 3.1 and 3.2. As a reader, my expectation after having the gLINEAR-$\kappa-RL problem defined was then to dive into the theorem and analysis around this problem. The flow is slightly disjointed by moving between the two problems, then going onto the analysis. But, I recognize this is personal taste, and I see the tidiness of the current framing as well. Just something to consider.
- For Theorem 3.1, the proof works for \epsilon_0 = 0.05. For this reason, I would state the theorem using 0.05 directly and not bother introducing epsilon_0.
- I can imagine it would be useful to motivate the choice of softmax policies a bit further. Or, at least to comment on whether you believe Theorem 3.2 is tailored to the choice of $\Pi^{sm}$, and to what extent you believe a similar/nearly-identical result will hold for most well-behaved policy classes.
- As you'll see, the abstract does not render well on openreview.

__Typos / Writing Suggestions__

Then, I include a few low-level reactions and suggestions below.

Abstract:
- You can safely remove this "(a variant of MAX-3SAT)", as the names of the problems makes this clear already.

Introduction:
- Supporting some of these statements with references would be useful: "...the estimation of the value function so that resulting complexity bounds become independent of the size of the state space"
- For Q2, the symbol "$\subsetneq$" seems unecessary here, rather than just $\subset$, right? Oh, I see, you want to specify that $\pi^*$ is in the set, but not the only policy in the set?
- Line 69: "by defining subclass" --> "by defining the subclass"
- Line 70: "of realizable policy set" --> "of realizable policy sets"
- Equation between lines 106/107: You use $\sup$ in all cases one but, where you move to the $\max$; any specific reason?

Preliminaries
- Line 114 header: "Linear Function Approximations" --> "Linear Function Approximation"

Problem Statement and Main Results
- Definition 3.3: I assume there is a good reason to make use of the states reachable after $h$ steps: $\mathcal{S}_h$, though it seems to add complexity for little benefit so far.
- Since the softmax policies do not use a temperature parameter, you might flag this somewhere early on. For instance, in the introduction, when you mention you will study softmax policies, you could add a short note indicating that there is no temperature used.
- Typo: "Theorem 3.1 (NP-Harness" --> "Theorem 3.1 (NP-Hardness"
- For Theorem 3.1: Is your intuition that the result also holds for values larger than 0.05, but that your proof simply won't generalize to those cases? Or, do you suspect for some larger epsilon that the problem becomes easier (this would be surprising)?

**Ethical Concerns:**

["NO or VERY MINOR ethics concerns only"]

**Final Justification:**

This is a very strong paper: it has a clear focus on a compelling theoretical question, and provides a clean and rigorous answer to this question.

No major issues were uncovered during the rebuttal, review, or discussion, so I maintain a positive outlook on the work.

**Limitations:**

Yes, the authors call out all assumptions and limitations.

**Quality:**

4

**Strengths And Weaknesses:**

__Strengths__
- [New, Interesting Problem] The relaxation of realizability to the partial case is a clever idea, and presents an interesting new problem.
- [Fundamental Result] The new hardness results are fundamental and interesting.
- [Focused Narrative] This paper is ultimately about one clean idea: partial $q^\pi$ realizability
- [Clarity] Despite the technical nature of the paper, it is very readable. A lot of work clearly went into making the ideas accessible.

__Weaknesses__
- There are no major weaknesses. For what this paper sets out to do, it executes on it to a high degree.
- Two minor weaknesses: (1) I could imagine clarifying some of the design choices in slightly more detail. For example, why did you choose the softmax policies to analyze, rather than another class? Do you anticipate similar results hold for most well-behaved choices of policy class? (2) The different feature vectors did leave me slightly confused by the end. Adding a little more exposition to help the reader understand this will help in communicating some of the main ideas.


In light of the above, unless another reviewer catches a glaring issue, I believe this is a clear accept.

---

> ### Author Rebuttal · Authors · 2025-07-31
>
> We sincerely thank the reviewer for their thoughtful and encouraging feedback, as well as for raising these insightful questions.
>
> ## Comment on Minor Weakness 1 (Selection of Softmax Policy and Anticipated Hardness Results Under General Policy Classes)
>
> We thank the reviewer for the thoughtful question. At the beginning of **Section 3.1 (“Problem Statement and Definitions”)**, in the part discussing softmax policies, we motivate our choice primarily due to the widespread use of such policies in actor-critic and policy gradient methods. Their effectiveness in promoting exploration and enabling smooth parameterization makes softmax a natural choice for our partial $q^{\pi}$-realizability setting.
>
> Regarding the second part of the question, we refer to **Remark 4.2**, which outlines the conditions necessary for achieving similar hardness results for general policy classes. In summary, an important requirement is that the policy set must be expressible in a parametric form (as is the case with greedy and softmax policies). Additionally, during the reduction process, it must be possible to construct any policy $\pi \in \Pi$ in polynomial time with respect to the input size of the given NP-hard problem. Together with the other conditions highlighted in Remark 4.2, these properties enable us to extend the hardness result to broader classes of policies.
>
> In future work, we plan to explore other well-defined policy classes, such as those discussed in [1], to examine whether the hardness result can still be maintained.
>
> ## Comment on Minor Weakness 2 (Having Two Feature Vectors)
>
> We appreciate the reviewer’s comment and the opportunity to clarify this point. The key reason for using different feature vectors is that it enables a cleaner and more tractable hardness proof, allowing us to bypass some otherwise very challenging and unnecessary technical complications.
>
> From a design perspective, overloading a single feature representation to simultaneously satisfy both the linear realizability condition and the policy class definition can significantly constrain the expressiveness of both components. This makes the theoretical and practical design considerably more difficult.
>
> Moreover, using two distinct representations aligns well with real-world scenarios. In practice, there is no requirement that the policy class available to the agent must be parameterized in the same way as the features used for learning. These are inherently separate concerns. One can easily imagine that the parameters for policy being executed come from a representation learning algorithm (e.g., via neural networks) or even from human expert demonstrations. Therefore, the main learning feature vector (i.e., the feature vector $\phi$) can—and often should—differ from the representation used to define the policy class. This separation provides flexibility and better models practical systems.
>
>
> ## Comment on Questions and Suggestions
> We sincerely thank the reviewer for the thoughtful and constructive high-level suggestions raised in the *Questions* section. We appreciate the positive assessment and take this opportunity to clarify several key points that shed more light on the motivation behind our framework and the technical details of our results.
>
>    - $$\textbf{[Suggestion regarding broader applicability of the hardness results]:}$$ We acknowledge that understanding how our hardness result interacts with different classes of policies and representations is a promising direction. In fact, the parametric design and constructibility conditions stated in Remark 4.2 can serve as a basis for testing other classes of policies, such as those studied in recent works like [1], and we are actively exploring this line of inquiry.
>    - $$ \textbf{[Typos / Writing Suggestions Section - Q2]:}$$ Your understading is completely true regarding this propoer subset notation, and we wanted to say that definitly optimal policy $\pi^* \in \Pi$, but along side of this policy other policies are also in the set.
>    - $$ \textbf{[Typos / Writing Suggestions Section - Equation between lines 106/107]:}$$ The use of $\sup$ instead of $\max$ is intentional and reflects a standard practice in analysis for ensuring generality. Specifically, when optimizing over the space of all policies $\pi$, the set of achievable value functions $v^{\pi}(s)$ may not have a maximum, even though a supremum exists. This situation typically arises in settings where:
>      - The space of policies is infinite or uncountable,
>      - The policy space is not compact, or
>      - There is no policy that attains the highest value due to the lack of continuity or closure in the space.
>
>
> In such cases, there may exist a sequence of policies $\pi_n$ for which $v^{\pi_n}(s)$ approaches the supremum, but no individual policy achieves the maximum. Hence, $\sup$ ensures that the value function $v^*(s) = \sup_{\pi} v^{\pi}(s)$ is always well-defined, even when an optimal policy does not exist in the strict sense. In finite MDPs, where the set of policies is finite or compact, the supremum is attained, and $\sup$ and $\max$ are equivalent. But to maintain mathematical rigor and generality, especially in theoretical settings, the supremum is preferred.
>
> - $$\textbf{[Typos / Writing Suggestions Section - Problem Statement and Main Results - Definition 3.3]:}$$ We use the notation $\mathcal{S}_h$ to denote the set of states reachable at step $h$ because, for any $s_h \in \mathcal{S}_h$, we associate a specific parameter vector $\theta_h$ and a feature vector $\phi(s_h, a_h)$. This horizon-dependent view of the state space is crucial for our hardness proof, particularly in the worst-case design of the learning approximation parameters: $\phi$, $\theta$, and the policy class parameterization $\phi'$. Including this structure makes the analysis and presentation of the proof clearer and more organized.
>
> - $$ \textbf{[Typos / Writing Suggestions Section - Problem Statement and Main Results - Theorem 3.1]:}$$
> We thank the reviewer for the thoughtful question. The threshold $\epsilon \leq 0.05$ in Theorem 3.1 is not fundamental to the inherent hardness of the underlying RL problem; rather, it arises from the specifics of our reduction. In particular, we reduce from the well-known NP-hard problem $\delta$MAX-3SAT, for which it is NP-hard to approximate within a factor of $\frac{7}{8}$. In our reduction, due to the scaling and normalization within the MDP construction, this gap translates into a value difference of at most $1/16$. Therefore, the theoretical limit for which our current reduction can prove hardness is $\epsilon \le 1/16$. The specific choice of $\epsilon = 0.05$ reflects a conservative parameter tuning to ensure clarity and correctness, but tighter analysis could likely push the bound closer to this $1/16$ threshold. Thus, we view the $\epsilon$-bound as a feature of the reduction and not as evidence that the problem becomes easier for larger $\epsilon$.
>
> We appreciate the reviewer’s insightful comments, which have helped us refine both the theoretical and practical positioning of our contributions. We will carefully address the noted typos and writing issues in the final version to ensure clarity and precision throughout the paper.
>
> **---[References]---**
>
> [1] Z. Jia, G. Li, A. Rakhlin, A. Sekhari, and N. Srebro. When is agnostic reinforcement learning statistically tractable?, 2023.

---

> > ### Comment · Reviewer_SdE3 · 2025-08-04
> > **Re: Rebuttal**
> >
> > I thank the authors for their thorough rebuttal, both to my review and all the reviewers. I have read through the reviews and rebuttals, and maintain that this is a very solid paper, and it should be accepted. The responses regarding the use of the softmax policy as the policy class make sense to me, and I would encourage the authors to add some of this justification to the text of the paper. Similarly, regarding the two feature vectors, this was a point a few reviewers picked up on and indicates it is something that can be clarified more in the paper. The separation the authors describe makes sense, but adding additional detail explaining this separation to the paper itself will help strengthen the narrative and bring more readers along.
> >
> > Overall, I believe this is a solid contribution, and I will happily argue for the paper to be accepted.

---

### Official Review · Reviewer_XQpi · 2025-06-30

**Clarity:** 3
**Significance:** 2
**Originality:** 3
**Rating:** 4
**Confidence:** 2

**Summary:**

Prior work has shown that RL is computationally tractable under the stronger $q^\pi$-realizability assumption, whereas the weaker $q^*$-realizability assumption leads to exponential lower bounds. To investigate whether improved results are achievable under assumptions weaker than $q^\pi$-realizability, this paper introduces a partial realizability assumption. It then considers two types of policy classes—greedy and softmax—and proves an exponential lower bound on the computational complexity in both cases.

**Questions:**

(1) The claims in Lines 214–219 seem a bit unclear to me. Specifically, the policies in $\Pi_g$ are deterministic, while those in $\Pi_{sm}$ are stochastic. Why should $\Pi_g \subseteq \Pi_{sm}$ hold with high probability? Also, over what is this probability defined?

(2) Theorem 3.2 does not explicitly claim that SLINEAR-κ-RL is NP-hard. This should perhaps be stated directly in the theorem.

**Ethical Concerns:**

["NO or VERY MINOR ethics concerns only"]

**Final Justification:**

After reading the rebuttal, I think it's a good paper. Thus, I lean towards acceptance.

**Limitations:**

Yes

**Quality:**

3

**Strengths And Weaknesses:**

Strengths:

The paper is well written, providing a clear introduction that conveys the intuition behind the problem and offering a detailed overview of the proof strategy. Although I did not verify every step of the proof, the theoretical analysis appears to be solid. The paper presents a novel result on the computational complexity of the RL problem, which, to the best of my knowledge, has not been previously reported in the literature.

---

Weaknesses:

(1) Definition 3.1 is somewhat ambiguous, as the weight parameter $\theta$ should include a superscript $\pi$ to indicate its dependence on the policy.

(2) One concern I have is that the result is entirely negative and does not hold for every policy class. In particular, because the feature map $\phi'$ is constructed as part of the proof, the result does not offer insight into which structural properties of a policy class might help reduce the problem’s complexity. Is there any example of a policy class with alternative feature representations that could potentially avoid this hardness? Furthermore, given that the computational hardness of RL under $Q^*$-realizability is already well-established, it is unclear to me what new understanding or contribution this paper adds relative to the existing body of work.

(3) I find the paper to be closely related to [1], especially in its approach to constructing an MDP from a SAT instance. However, it lacks a comprehensive comparison of its proof techniques with those of [1] and other relevant prior works.

[1] Exponential Hardness of Reinforcement Learning with Linear Function Approximation, Kane et al. 2023

---

> ### Author Rebuttal · Authors · 2025-07-31
>
> We thank the reviewer for their valuable comments and constructive feedback. We appreciate your time in reading the paper carefully and for raising important points. Below, we address your main concerns in detail.
>
> ## Comment on Weakness 1 (Ambiguity of Definition 3.1)
>
> We agree with the reviewer that, technically, the parameter $\theta$ depends on the policy. However, in much of the literature, this dependency is often left implicit to improve the clarity and readability of expressions. Since the policy class is indexed by $\theta$, it is generally understood that $\theta$ parameterizes the policy, and adding a superscript to denote this dependence is redundant in context. That said, we acknowledge the potential ambiguity and will clarify this point in the revised version.
>
> ## Comment on Weakness 2 (Clarification of Our Contributions)
>
> We appreciate this insightful observation. First, note that this is a ***hardness result***, so it is not necessary to demonstrate NP-hardness for all possible policy classes. It suffices to show that ***there exists*** a policy class for which the learning problem becomes NP-hard. In our work, we show that two broad and widely used policy classes---greedy and softmax policies---indeed have this property.
>
> Importantly, this work does not aim to investigate structural properties of policy classes. Our focus is solely on the fact that the policy class contains the optimal policy along with some additional policies, which introduces a new family of problems characterized by partial $q^{\pi}$-realizability.
>
> The concerns raised in the comment are certainly relevant when the goal is to design efficient algorithms---statistically or computationally. In such settings, one may leverage structural assumptions on the policy class to guide the design of efficient algorithms. However, that is not the objective of this paper.
>
> In the $q^*$-realizability setting, the policy class contains only the optimal policy, and prior work has already established hardness results for this case. In contrast, our goal is to examine how the hardness result evolves when we expand the policy set to include other policies as well---thus moving toward the stronger assumption of $q^{\pi}$-realizability, under which efficient algorithms are known to exist. Our result provides insight into the limits of algorithm design, by showing that even under rich policy classes such as greedy and softmax, one cannot hope for efficient solutions in general.
>
> We acknowledge that establishing hardness results for other types of policy classes would pose a valuable technical challenge. However, such an extension is not essential to our main message, as the policy classes we study already encompass two of the most widely used classes in practical reinforcement learning algorithms, namely argmax and softmax.
>
> ## Comment on Weakness 3 (Comprehensive Comparison of Proof Techniques)
> We thank the reviewer for this insightful comment. The referenced work [1] is indeed one of the key related papers, and we have discussed its connection to our approach in detail in the *Extended Discussion of Related Work* section in the appendix. While both papers reduce from SAT-style problems, the settings and techniques differ significantly. In [1], the policy class includes only a single deterministic optimal policy, and the proof techniques are tailored to that restricted setting. In contrast, our work considers more general and potentially stochastic policy classes (e.g., softmax policies), which necessitates a randomized reduction technique to reflect this broader structure. Furthermore, our MDP construction techniques are novel and differ substantially from those in [1], as they are specifically designed to accommodate the partial $q^\pi$-realizability setting with a richer policy class. These distinctions are critical for achieving our hardness result and are explained in detail in the appendix. To clarify this point, we will bring some of these discussions to the main body of the paper in the revised version. Thank you again for the comment.
>
> ## Answer to Question 1 (Clarification of $\Pi_g \subseteq \Pi_{sm}$)
>
> Thank you for the question. In our construction, the softmax policy class $\Pi_{sm}$ is parameterized by $\theta' \in \mathbb{R}^{d'}$. By appropriately choosing values of $\theta'$, the softmax policy can assign arbitrarily high probability to a specific action in each state. Consequently, $\Pi_{sm}$ can approximate any deterministic greedy policy arbitrarily closely. This is what we mean by stating that "$\Pi_g \subseteq \Pi_{sm}$ holds with high probability."
>
> To be more formal, we appeal to an analogy with the coupon collector problem. Suppose there are $N$ distinct deterministic greedy policies in $\Pi_g$, each corresponding to a different configuration of preferred actions across states. If we randomly sample $O(N \log N)$ parameter vectors $\theta'$ (from a distribution with full support over $\mathbb{R}^{d'}$), then with high probability, for each greedy policy in $\Pi_g$, there exists a sampled $\theta'$ such that the resulting softmax policy assigns sufficiently high probability to the same action choices—effectively mimicking that greedy policy.
>
> Thus, the probability is defined over the random draws of $\theta'$ used to generate policies in $\Pi_{sm}$. Because the parameter space is continuous and unbounded, we can sample enough softmax policies to cover all greedy policies with high probability.
>
> We will clarify this interpretation further in the final version to improve precision and readability.
>
> ## Answer to Question 2 (NP-Hardness of SLINEAR-κ-RL in Theorem 3.2)
>
> Thank you for your comment. You are correct that Theorem 3.2 does not explicitly state that SLINEAR-$\kappa$-RL is NP-hard. However, the theorem is based on the Randomized Exponential Time Hypothesis (rETH), which is strictly stronger than the standard assumption that P $\ne$ NP. Specifically, rETH rules out not only polynomial-time deterministic and randomized algorithms, but also sub-exponential time randomized algorithms for NP-complete problems like 3-SAT.
>
> Therefore, the intractability result in Theorem 3.2 already implies NP-hardness, and in fact goes further by showing that no sub-exponential time randomized algorithm (with low error probability) exists for SLINEAR-$\kappa$-RL under rETH.
>
> **--- [References]---**
>
> [1] D. Kane, S. Liu, S. Lovett, G. Mahajan, C. Szepesv´ari, and G. Weisz. Exponential hardness of reinforcement learning with linear function approximation, 2023.

---

> > ### Comment · Reviewer_XQpi · 2025-08-03
> > **Response to authors**
> >
> > Thanks for your clarification.
> >
> > The response addresses most of my concerns, and I believe the work is overall solid. However, I still retain some minor concerns related to Weakness 2. While I understand that structural assumptions on the policy class are not the main focus of this paper, which is acceptable. I believe that, given the existing positive and negative results at the two extremes, a positive result under an intermediate assumption would carry more significance than a negative one.
> >
> > Regarding my specific questions, I do not believe the paper contains major errors. However, some claims, such as the one involving the subset relation ($\subseteq$), are not stated with sufficient rigor. I encourage the authors to revise these points for the final version.
> >
> > Overall, I think this is a good paper, and I will maintain my current score.

---

### Official Review · Reviewer_FVp2 · 2025-07-03

**Clarity:** 2
**Significance:** 3
**Originality:** 3
**Rating:** 4
**Confidence:** 1

**Summary:**

This paper studies the computational complexity of reinforcement learning under linear function approximation. The authors introduce a new notion, partial $q^{\pi}$-realizability, which lies between the existing assumptions of $q^*$-realizability and $q^{\pi}$-realizability. They show that finding an $\varepsilon$-optimal policy within the greedy policy class is NP-hard, via a polynomial-time reduction from $\delta$-MAX-3SAT. For the softmax policy class, they establish an exponential time lower bound in the feature dimension under the randomized Exponential Time Hypothesis (rETH). The authors conclude that even when the policy class is relaxed beyond the optimal policy, the computational hardness of reinforcement learning remains unchanged.

**Questions:**

1. Under the *partial* $q^\pi$-realizability assumption, what additional conditions would allow for efficient learning?
2. Although the paper is theoretical in nature, it would be more convincing if a small illustrative MDP example demonstrated the computational failure or blow-up of existing linear function approximation algorithms. Is there a simple experiment that could be conducted?
3. Could similar lower bounds be shown under weaker access assumptions than the generative model? If not, could you explain what technical challenges prevent this?

**Ethical Concerns:**

["NO or VERY MINOR ethics concerns only"]

**Final Justification:**

I believe this paper has met the bar for acceptance, and I will maintain my original assessment.

**Limitations:**

yes

**Paper Formatting Concerns:**

I did not find any significant formatting issues in the paper.

**Quality:**

3

**Strengths And Weaknesses:**

**Strengths**

The paper proposes a novel realizability assumption called *Partial* $q^\pi$-realizability, which bridges the gap between the existing $q^*$- and $q^\pi$-realizability assumptions. It establishes fundamental computational hardness results under linear function approximation by leveraging different complexity hypotheses for two distinct policy classes. The logical structure of the paper is clearly organized, making it easy to follow.


**Weaknesses**


As acknowledged in the paper’s limitations, it remains unaddressed whether similar lower bounds hold in the case of shared features.

---

> ### Author Rebuttal · Authors · 2025-07-31
>
> We sincerely thank the reviewer for the thoughtful and constructive feedback. We are encouraged by the engagement with our work and appreciate the opportunity to clarify key aspects of our approach, assumptions, and contributions. Below, we address each of the reviewer’s comments in detail.
>
> ## Comment on Weakness 1 (The Use of Separate Feature Representations)
> We appreciate the reviewer’s comment regarding the use of two different feature representations. As clarified in the paper, our problem formulation involves two feature vectors: $\phi'$ is used solely for defining the policy class, while $\phi$ is used for learning. This separation enables us to construct rich policy classes without the restriction that the same representation must satisfy the linear realizability condition, which is required only for $\phi$.
> In contrast, if we were restricted to using a single feature vector for both policy class construction and learning, we would face significant technical challenges. Specifically, any constraint imposed by the policy class would directly affect the realizability condition, making the design of the linear approximation architecture highly constrained and difficult to analyze.
> This modeling choice is also motivated by real-world applications where the policy class may come from external sources — for example, from human demonstrations or a neural policy prior — and is thus independent of the features used in learning. A concrete example is autonomous driving, where the set of feasible trajectories (i.e., the policy class) may be determined by physical parameters such as engine power or wheel dynamics, while the learning features could be derived from GPS-based localization, camera input, or LIDAR — representations not necessarily tied to the internal mechanics of the vehicle.
> That said, we agree that understanding the complexity of learning under a single feature vector remains an important technical question. We experimented with constructions such as concatenating $\phi$ and $\phi'$ into a joint representation, but found that this approach introduces new complications due to our hardness construction methods. We leave this as an open and important direction for future work.
>
> ## Answer to Question 1 (Additional Conditions for Efficient Learning)
>
> We interpret "efficient learning" here as referring to computational efficiency. This is indeed a compelling and largely open problem. Existing works have primarily focused on the extremes of full $q^*$-realizability and full $q^\pi$-realizability, and relatively no attention has been paid to intermediate cases like the partial $q^\pi$-realizability setting we introduce.
> One clear case where efficient learning is possible is when the policy class includes all possible policies — in such cases, previous results (e.g., those assuming full coverage with generative model access) imply tractability. However, beyond this trivial case, it remains unclear what structural properties of the policy class or function class (e.g., covering number, spanning capacity) would ensure computational tractability under partial realizability.
> On the statistical side, we believe that results from the Agnostic RL literature [1] may offer useful tools. In particular, the notion of bounded spanning capacity in that work might extend to our setting with linear function approximation, suggesting a potential path toward identifying conditions for statistical efficiency.
> Exploring both the computational and statistical aspects of learning under partial realizability is part of our ongoing work.
>
> ## Answer to Question 2 (Possibility of Simple Experiment)
> We appreciate this suggestion. However, current linear function approximation algorithms are typically designed for $q^*$-realizability or full $q^\pi$-realizability settings. These assumptions are fundamentally incompatible with our partial $q^\pi$-realizability setting, making such algorithms inapplicable without significant modification. As a result, any empirical failure of these methods on our constructed MDPs would primarily reflect an assumption mismatch rather than illuminate the computational hardness at the core of our contribution.
> That said, an important direction for future work is to investigate how existing algorithms can be adapted or extended to work under partial $q^\pi$-realizability. This would help bridge the gap between theoretical hardness results and practical algorithmic design. Once such adaptations are available, they could serve as meaningful baselines to empirically test the computational challenges in this regime.
>
> ## Answer to Question 3 (Hardness Results under Weaker Access Assumptions)
>
> Yes — our hardness results extend to weaker access models, including the online access model. In fact, our lower bound construction remains valid under this assumption, and the proof does not require modification. The online model restricts learners to follow complete trajectories rather than arbitrarily querying any state-action pair, making it a strictly weaker assumption than generative model access. Thus, our hardness results apply directly to this setting as well.
>
> **--- [References]---**
>
> [1] Z. Jia, G. Li, A. Rakhlin, A. Sekhari, and N. Srebro. When is agnostic reinforcement learning statistically tractable?, 2023.

---

> > ### Comment · Reviewer_FVp2 · 2025-08-06
> >
> > I appreciate the authors’ detailed response to my questions. I agree that the authors did solid work, and I would like to participate in the discussion with the other reviewers before finalizing my score.

---

### Note · Authors · 2025-08-13

We sincerely thank all the reviewers for their insightful and constructive feedback. We will incorporate revisions in the final version of the paper, particularly:
1. Clarifying the motivation for using dual feature vectors, as suggested by **FVp2**, **SdE3**, and **v2tA**, and explaining how a practical example such as the autonomous driving scenario mentioned during the rebuttal phase connects to this choice.
2. Providing additional explanation regarding the reasoning behind $\Pi^g \subset \Pi^{sm}$, as suggested by **XQpi**.
3. Refining the overall writing and presentation of the paper, including addressing the typos and suggestions provided by **SdE3**.

We once again thank all the reviewers for their thoughtful and constructive input.

---

### Decision · Program_Chairs · 2025-09-17

**Decision:**

Accept (poster)

**Comment:**

This paper studies the computational hardness of RL with a generative model under the so-called partial $q^{\pi}$ realizability, where there is a subset of policies under which $q^{\pi}$ is linearly-realizable for all $\pi$ in the subset. They consider two types of policy subset: greedy policies and softmax polices. They establish NP hardness for greedy policies and exponential lower bounds for softmax policies.


**Strengths**. The partial $q^{\pi}$ realizability is well motivated that sits between two other regimes that are more well studied: $q^*$-realizability and all-policy $q^{\pi}$ realizability. The paper is well written and provides a comprehensive related work discussion as well as the high-level ideas of their construction approach. The hardness problem $\delta$-MAX-3SAT seems a novel technical construction compared to other hardness problem constructions in the growing literature of hardness of RL. Understanding computational complexity in RL is a significant problem for the ML community.

**Weaknesses**. No major issues, as agreed by the reviewers.

This paper receives positive evaluations from all the reviewers, initially and throughout the discussion. In particular, the discussion phase presents a collegial discussion where the authors have well addressed the issues (mostly minor) raised by the reviewers and the reviewers well acknowledged such responses.